Cenozoic aridization in Central Eurasia shaped diversification of toad-headed agamas (Phrynocephalus; Agamidae, Reptilia)

Solovyeva Evgeniya N. 1 anolis@yandex.ru
Lebedev Vladimir S. 1
Dunayev Evgeniy A. 1
Nazarov Roman A. 1
Bannikova Anna A. 2
Che Jing 3 4
Murphy Robert W. 3 5
http://orcid.org/0000-0002-7576-2283 Poyarkov Nikolay A. 2 n.poyarkov@gmail.com
1 Zoological Museum, Lomonosov Moscow State University , Moscow , Russia
2 Biological Faculty, Department of Vertebrate Zoology, Lomonosov Moscow State University , Moscow , Russia
3 State Key Laboratory of Genetic Resources and Evolution, and Center for Excellence in Animal Evolution and Genetics, Kunming Institute of Zoology, Chinese Academy of Sciences , Kunming, Yunnan , China
4 Southeast Asia Biodiversity Research Institute, Chinese Academy of Sciences , Yezin, Nay Pyi Taw , Myanmar
5 Faculty of Arts and Science, Department of Ecology & Evolutionary Biology, University of Toronto , Toronto, ON , Canada
Pie Marcio
Electronic publication date: 2018 Mar 19
Publication date: 2018
Volume: 6
Electronic Location ID: e4543
Received 2017 Nov 24; Accepted 2018 Mar 6
Copyright: © 2018 Solovyeva et al.
Copyright year: 2018
Copyright holder: Solovyeva et al.
License: This is an open access article distributed under the terms of the Creative Commons Attribution License, which permits unrestricted use, distribution, reproduction and adaptation in any medium and for any purpose provided that it is properly attributed. For attribution, the original author(s), title, publication source (PeerJ) and either DOI or URL of the article must be cited.
License URL: https://creativecommons.org/licenses/by/4.0/

Keywords: Squamata, Reptilia, Dispersal–extinction–cladogenesis, Sauria, Agamidae, Asian deserts, Tectonics, Biogeography, Evolution, Palearctic, Mid-Miocene climate transition, Himalayan uplift

Funding: Russian Foundation of Basic Research (Grant Nos. RFBR 15-04-08393 and RFBR 15-29-02771.Strategic Priority Research Program (A) (Tibet program) of the Chinese Academy of Sciences (CAS) Animal Branch of the Germplasm Bank of Wild Species, CAS (Large Research Infrastructure Funding) Russian Science Foundation 14-50-00029 This work was supported by the Russian Foundation of Basic Research (Grant Nos. RFBR 15-04-08393 and RFBR 15-29-02771) to Nikolay A. Poyarkov (molecular experiments, phylogenetic analyses) and by Strategic Priority Research Program (A) (Tibet program) of the Chinese Academy of Sciences (CAS), and the Animal Branch of the Germplasm Bank of Wild Species, CAS (Large Research Infrastructure Funding) to Jing Che. Specimen storage and examination was completed with financial support of Russian Science Foundation (RSF grant No. 14-50-00029). The funders had no role in study design, data collection and analysis, decision to publish, or preparation of the manuscript.

==============================
We hypothesize the phylogenetic relationships of the agamid genus Phrynocephalus to assess how past environmental changes shaped the evolutionary and biogeographic history of these lizards and especially the impact of paleogeography and climatic factors. Phrynocephalus is one of the most diverse and taxonomically confusing lizard genera. As a key element of Palearctic deserts, it serves as a promising model for studies of historical biogeography and formation of arid habitats in Eurasia. We used 51 samples representing 33 of 40 recognized species of Phrynocephalus covering all major areas of the genus. Molecular data included four mtDNA (COI, ND2, ND4, Cytb; 2,703 bp) and four nuDNA protein-coding genes (RAG1, BDNF, AKAP9, NKTR; 4,188 bp). AU-tests were implemented to test for significant differences between mtDNA- and nuDNA-based topologies. A time-calibrated phylogeny was estimated using a Bayesian relaxed molecular clock with nine fossil calibrations. We reconstructed the ancestral area of origin, biogeographic scenarios, body size, and the evolution of habitat preference. Phylogenetic analyses of nuDNA genes recovered a well-resolved and supported topology. Analyses detected significant discordance with the less-supported mtDNA genealogy. The position of Phrynocephalus mystaceus conflicted greatly between the two datasets. MtDNA introgression due to ancient hybridization best explained this result. Monophyletic Phrynocephalus contained three main clades: (I) oviparous species from south-western and Middle Asia; (II) viviparous species of Qinghai–Tibetan Plateau (QTP); and (III) oviparous species of the Caspian Basin, Middle and Central Asia. Phrynocephalus originated in late Oligocene (26.9 Ma) and modern species diversified during the middle Miocene (14.8–13.5 Ma). The reconstruction of ancestral areas indicated that Phrynocephalus originated in Middle East–southern Middle Asia. Body size miniaturization likely occurred early in the history of Phrynocephalus. The common ancestor of Phrynocephalus probably preferred sandy substrates with the inclusion of clay or gravel. The time of Agaminae radiation and origin of Phrynocephalus in the late Oligocene significantly precedes the landbridge between Afro-Arabia and Eurasia in the Early Miocene. Diversification of Phrynocephalus coincides well with the mid-Miocene climatic transition when a rapid cooling of climate drove progressing aridification and the Paratethys salinity crisis. These factors likely triggered the spreading of desert habitats in Central Eurasia, which Phrynocephalus occupied. The origin of the viviparous Tibetan clade has been associated traditionally with uplifting of the QTP; however, further studies are needed to confirm this. Progressing late Miocene aridification, the decrease of the Paratethys Basin, orogenesis, and Plio–Pleistocene climate oscillations likely promoted further diversification within Phrynocephalus. We discuss Phrynocephalus taxonomy in scope of the new analyses.

Introduction

Historical biogeography aims to understand the drivers of speciation including the roles played by plate tectonics and climatic change (Lomolino et al., 2006). The eastern part of the Great Palearctic Desert Belt spans from Eastern Europe to Eastern China, including Middle Asia (Kazakhstan, Kyrgyzstan, Uzbekistan, Tajikistan, and Turkmenistan) and Central Asia. Middle and Central Asia have one of the oldest desert areas. Desertification started at least 23.8–22.0 million years ago (Ma) (Xia & Hu, 1993; Guo et al., 2002). Various paleogeographic factors played major roles in the shifting of Central Eurasian climate (Ramstein et al., 1997). These include the Miocene retreat of the Paratethys Sea, which stretched over Eurasia 30 Ma (Popov et al., 2004, 2009), tectonic activity in Southwest Asia (Whiteman, 1978; Weise, 1974; Macey et al., 1993; Golonka, 2004), and the uplifting of the Qinghai–Tibetan Plateau (QTP; Harrison et al., 1992, 1995; Ramstein et al., 1997; Zhisheng et al., 2001; Molnar, 2005). Aridization led to the disappearance of forests and formation of desert ecosystems (Cerling et al., 1997; Ma et al., 1998) and it intensified in the late Cenozoic following the formation of Asian monsoon climate (Guo et al., 2002).

Central Eurasian deserts cradle a rich herpetofauna (Chernov, 1948, 1959; Likhnova, 1992; Ananjeva & Tuniev, 1992; Szczerbak, 2003). In the late Cenozoic, dramatic climatic changes influenced the origins, diversification and distribution of Central Eurasian reptiles (Macey et al., 2000; Melville et al., 2009). However, the dearth of phylogenetic and historical biogeographic studies for Central Eurasia does not allow the testing of hypotheses on the biological consequences of Cenozoic climatic events. The reptile fauna of the Central Asian deserts is particularly diverse, yet we still have limited understanding of the drivers of evolution of the constituent species (Melville et al., 2009).

The agamid genus Phrynocephalus Kaup, 1825, or toad-headed lizards, is one of the most speciose genera in its family. It contains from 28 to over 42 species and spans arid regions from northwestern China to the western side of the Caspian Sea, across the QTP, and Southwest Asia to the Arabian Peninsula (Wermuth, 1967; Moody, 1980; Barabanov & Ananjeva, 2007; Guo & Wang, 2007; Uetz & Hošek, 2016; Kamali & Anderson, 2015) (Fig. 1). The species are ecologically important components of the Central Eurasian desert fauna and are highly adapted to sand dunes and stony montane deserts from sea level up to 6,400 m a.s.l. (Zhao, Zhao & Zhou, 1999). They exhibit high levels of variation in ecological and morphological diversity, and the species range from being habitat generalists to specialists (Clemann et al., 2008; Dunayev, 2009). Oviparous reproduction occurs in lower elevations and yet viviparous species occur on the QTP (Zhao & Adler, 1993; Pang et al., 2003; Guo & Wang, 2007). The involvement of Phrynocephalus in so-called “substrate races” leads to much taxonomic confusion (Dunayev, 2009), especially because their phylogenetic relationships and historical biogeography remain uncertain. Considerable taxonomic, morphological, allozyme, karyological, osteological, and ethological research has been conducted on the charismatic Phrynocephalus of Central Asia (for a brief review on history of phylogenetic studies of the genus Phrynocephalus see Supplemental Information 1). Regardless the phylogenetic and taxonomic relationships within the toad-headed agamas remain controversial and largely unresolved (Ananjeva & Tuniev, 1992; Arnold, 1999; Macey et al., 1993; Dunayev, 1996b; Golubev, 1993; Zhao & Adler, 1993; Pang et al., 2003; Ananjeva et al., 2006; Solovyeva et al., 2011, Solovyeva, Dunayev & Poyarkov, 2012; and references therein). Hypothesis-testing can help deduce their origin, diversification and dispersal (Guo & Wang, 2007; Melville et al., 2009; Solovyeva et al., 2014). The most complete genealogic hypothesis obtained up to date (Solovyeva et al., 2014) is based entirely on the mtDNA data; several major nodes of the tree have little or no support, so elaboration of a more robust phylogeny based on nuclear markers is needed.

Figure 1 Current distribution and species richness of the genus Phrynocephalus.

Color indicates the number of sympatric species of Phrynocephalus (from one to over four).

Herein, we explore a number of unresolved questions by using both mitochondrial and nuclear DNA markers based for 36 species of Phrynocephalus that cover the entire range of the genus. Specifically, we pursue three main objectives: (1) test the hypothesis that the nuDNA and mtDNA trees give compatible estimations of historical relationships; (2) evaluate hypotheses concerning potential climatic and tectonic drivers of speciation by using time-tree ages of each lineage calibrated based on molecular dating and fossils; and (3) reconstruct the ancestral distributions to differentiate among competing scenarios of historical biogeography. Our work offers the most complete taxon sampling to date by including up to 70% of the diversity of the genus. It helps to resolve longstanding phylogenetic and biogeographic issues of Central Eurasian biogeography and provides insights into the biogeographic consequences of Cenozoic aridization.

Materials and Methods

DNA samples

We used 51 samples representing 33 nominal species of Phrynocephalus from the collection of Zoological Museum of Moscow University (ZMMU). The primary outgroup included six other Agaminae species from the genera Agama Daudin, 1802, Paralaudakia Baig, Wagner, Ananjeva & Böhme, 2012, Stellagama Baig, Wagner, Ananjeva & Böhme, 2012, and Trapelus Cuvier, 1817 (Tables S1 and S2). For alcohol-preserved voucher specimens, muscle tissue was removed and preserved in 96% ethanol and stored subsequently at −35 °C; three tissue samples were obtained from the dried skin of voucher specimens.

DNA extraction, PCR conditions, and sequencing

Muscle and skin tissues were digested with Proteinase K and total genomic DNA was extracted using a standard phenol–chloroform extraction protocol followed with ethanol precipitation of DNA (Sambrook, Fritsch & Maniatis, 1989). Our analyses used the mitochondrial DNA dataset of Solovyeva et al. (2014), which included the following four mtDNA gene fragments: 654 bp of COI (cytochrome oxidase subunit I), 1,053 bp of ND2 (NADH-dehydrogenase subunit II), 705 bp of ND4 (NADH-dehydrogenase subunit IV), and 297 bp of Cytb (cytochrome b). The total length of the concatenated mtDNA genes was 2,703 bp (Table S3). We also amplified exons of four nuclear DNA genes as follows: 1,455 bp of RAG-1 (recombination activating gene), 675 bp of BDNF (brain derived neurotrophic factor), 1,182 bp of AKAP9 (A-kinase anchor protein 9), and 876 bp of NKTR (natural killer-tumor recognition). The total length of these data was 4,188 bp (Table S3).

Primer pairs for PCR were taken from the literature (mtDNA: Ivanova, DeWaard & Hebert, 2006; Wang & Fu, 2004; Macey et al., 2000; Arévalo, Davis & Sites, 1994; Pang et al., 2003; nuDNA: Shoo et al., 2008; Townsend et al., 2008, 2011) or designed by us (Table S4). PCR amplifications were performed in a reaction volume of 20 μl containing ca. 100 ng of template DNA, 0.3 pM/μl of each PCR primer, 1xTaq-buffer containing 25 mM MgCl2 (Silex, Moscow, Russia), 0.2 mM dNTPs, and 1 unit of Тaq-polymerase (Silex, Moscow, Russia; 5 units/μl). Protocols for PCR amplification were provided in the Supplemental Information 2. PCR products were purified with alcohol precipitation and a PCR purification kit (Isogen, Moscow, Russia). Purified products were sequenced with both forward and reverse primers using ABI PRISM® BigDye™ Terminator v.3.1 reagents and an Applied Biosystems 3730 DNA Analyzer (Applied Biosystems, Carlsbad, CA, USA). All sequencing followed the manufacturer’s protocols as given in the Engelgart’s IMB RAN (Moscow, Russia). All unique sequences were deposited in GenBank (Tables S1 and S2).

Taxa selection and molecular data

We added six sequences of Phrynocephalus available in GenBank to the final alignments (Table S2). Thirty-seven agamid species were selected as outgroup taxa for phylogenetic inference and time-tree calibration. These included the following Near Eastern and Middle Asian genera closely related to Phrynocephalus (Agaminae: Paralaudakia, Laudakia Gray, 1845, Trapelus, Stellagama) as well as more distant Southeast Asian agamids (Draconinae: Acanthosaura Gray, 1831, Draco Linnaeus, 1758, Calotes Daudin, 1802; Leiolepidinae: Leiolepis Cuvier, 1829) and Australian taxa (Amphibolurinae: Moloch Gray, 1841, Pogona Storr, 1982, Chlamydosaurus Gray, 1825). The most distant outgroup taxa also included representatives of Chamaeleonidae, Phrynosauridae, Dactyloidae, Iguanidae, Corytophanidae, Tropiduridae, Polychrotidae, Leiocephalidae, Lacertidae, Opluridae, Crotaphytidae, including representative taxa of the following agamid genera: Stellagama, Trapelus, Paralaudakia, and Agama for both nuclear and mtDNA dataset, and additionally representative taxa of the genera Xenagama Boulenger, 1895, Laudakia, Bufoniceps Arnold, 1992, Pseudotrapelus Fitzinger, 1843 and Calotes for the mtDNA dataset. Details on taxonomy, GenBank accession numbers and associated references were summarized in Tables S1 and S2.

Phylogenetic inference

Sequences were first aligned using the Clustal W algorithm (Thompson, Higgins & Gibson, 1994) in BioEdit Sequence Alignment Editor 7.1.3.0 (Hall, 1999), with default parameters. Subsequently, the alignment was checked and manually revised if necessary using Seqman 5.06 (Burland, 1999). Genetic distances were calculated using MEGA 6.1 (Tamura et al., 2013).

Phylogenetic tree reconstructions were performed with the following data sets: (1) each nuclear gene separately; (2) all nuclear genes concatenated; (3) all nuclear genes combined in a species-tree estimation; (4) a concatenation of four mitochondrial genes as in Solovyeva et al. (2014) but with the addition of Phrynocephalus rossikowi Nikolsky, 1898. To test whether the inclusion of distant outgroups can introduce any bias into results of tree inference, the nuclear and mitochondrial concatenations were put through an additional set of reconstructions omitting all non-agamid and non-agamine taxa, respectively. The optimum partitioning schemes for nuclear and mitochondrial alignments were identified with PartitionFinder (Lanfear et al., 2012) using greedy search algorithm under the AICc criterion.

Phylogenetic trees were reconstructed under the maximum likelihood (ML), maximum parsimony (MP), and Bayesian inference (BI) criteria. The ML trees were generated in Treefinder v.March 2011 (Jobb, 2011). For each subset, the best fitting substitution model was selected using the Bayesian Information Criterion in Treefinder. Nodal support was assessed by 1,000 bootstrap replications (BSP) and expected likelihood weights (ELW). The unweighted MP analyses were conducted in PAUP* v4.0b10 (Swofford, 2002) with 1,000 bootstrap replications. Bayesian inference was performed in MrBayes v3.1.2 (Ronquist & Huelsenbeck, 2003) with two simultaneous runs, each with four chains, for 200 million generations. We checked the convergence of the runs and that the effective sample sizes were all above 200 by exploring the likelihood plots using TRACER v1.5 (Rambaut & Drummond, 2007). The initial 10% of trees were discarded as burn-in. Confidence in tree topology was assessed by posterior probability (BPP) (Huelsenbeck & Ronquist, 2001).

Species-tree estimation was performed in *BEAST (Heled & Drummond, 2010) using the four independent nuclear loci. Prior to the analysis, the molecular clock assumption was tested separately for each exon by hierarchical likelihood ratio tests using PAML v4.7 (Yang, 2007). Following the results of these tests, we used a strict clock model for BDNF and uncorrelated lognormal relaxed clock models for the other three loci. No calibration information was utilized; the clock rate for BDNF was set to one. We used the same models and partitioning scheme as in the ML analysis. A Yule prior for the species-tree shape and the piecewise constant population size model were assumed. Default priors were used for all other parameters. Two runs of 500 million generations were conducted in BEAST v1.8.0 (Drummond et al., 2012). Parameter convergence was assessed in Tracer; the first 10% of generations were discarded as the burn-in. TreeAnnotator v1.8.0 (part of the BEAST package) was used to generate the maximum clade credibility tree.

Partition homogeneity test (Farris et al., 1994, 1995) as implemented in PAUP* v4.0b10 (Swofford, 2002) was used to ensure the absence of significant conflict among the four nuclear datasets (p-value = 0.071). We a priori regarded tree nodes with BSP values 75% or greater and BPP values over 0.95 as sufficiently resolved (Felsenstein, 2004; Huelsenbeck & Hillis, 1993). BSP values between 75% and 50% (BPP between 0.95 and 0.90) were regarded as tendencies and below 50% (BPP below 0.90) were considered to be not well-supported.

Congruence between nuclear and matrilineal genealogy

We tested if the mitochondrial genealogy of Solovyeva et al. (2014) was compatible with our nuDNA phylogeny to eliminate the possibility of mito-nuclear discordance and an introgressed mitogenome. ML trees with unconstrained and alternative constrained topologies were generated for the mitochondrial and nuclear datasets by using Treefinder v.March 2011. Treefinder was also used to calculate site-wise log-likelihoods and to perform the approximately unbiased tree-selection test (AU; Shimodaira, 2002). Significant discordance would have precluded a total evidence approach that evaluated together the mtDNA and nuDNA datasets because we wanted to differentiate between the initial cladogenic event(s) and the timing of interspecific hybridization(s), if present.

Divergence time estimates

The mtDNA dataset of Solovyeva et al. (2014) and our nuDNA concatenation were used to define divergence times in BEAST v1.8.0 (Drummond et al., 2012). Site and clock models were set as in the species-tree reconstruction. Analyses were run for 100 million of generations and the Yule model was set as the tree prior. Because no reliable paleontological data have been reported for Phrynocephalus, we used ten fossils from non-agamid outgroup taxa and outgroup Agamidae as calibration points (see Table S5; Fig. S1).

Area delimitation and biogeographic reconstruction

We used the ML of Lagrange (Ree et al., 2005; Ree & Smith, 2008) to reconstruct the biogeographic history of Phrynocephalus. Transitions between discrete states (ranges) along tree branches were modeled as a function of time, thus enabling ML estimation of the ancestral states at cladogenic events. Lagrange found the most likely ancestral areas at a node, the split of the areas in the two descendant lineages, and calculated the probabilities of these most likely areas at each node (Ree & Smith, 2008). We defined seven regions for the analyses: Kazakhstan, North Caspian and Ciscaucasian deserts (KZ), Central Asia (CA), Minor Asia and Transcaucasia (MI), Tibet (TI), Turan (TU), Middle East (ME), and Near East and Arabia (AR) (for details on biogeographic regions definition and references see Supplemental Information 3). The maximum number of regions included in one area was limited to two. We set two periods of time: before 10 Ma and after 10 Ma. This date echoed the considerable uplifting of the Pamir, Tianshan, and Karakoram mountains. (Abdrakhmatov et al., 1996). The matrices of the modern distribution areas were given in Table S6.

We reconstructed ancestral substrate niche evolution in Phrynocephalus under the MP criterion using MPRsets command in PAUP* v4.0b10 (Swofford, 2002) based on nuDNA topology with outgroup taxa included or excluded from the analysis. Polytomies in the nuDNA-based tree were resolved in accordance with the mtDNA topology. To account for topological uncertainty, the analysis was repeated based on a tree sample (180) from the posterior distribution produced by BEAST. Substrate niche was coded using six character states: (1) loose sand dunes; (2) fixed sands mixed with clay or gravel; (3) gravel and stone deserts; (4) clay soils and salines; (5) clay soils mixed with gravel; and (6) large rocks and cliffs. Transitions between states were formalized using step-matrix (Table S7).

To examine the evolution of body size in Agaminae, we used weighted squared-change parsimony (Maddison, 1991) as implemented in Mesquite v3.31 (Maddison & Maddison, 2017). We tested maximum SVL of taxa reported in literature or based on examination of voucher specimens. Maximum SVLs for each taxon were provided in Table S6.

Results

Taxon sampling, data collection, and sequence characteristics

The complete, aligned matrix contained 38 samples of Phrynocephalus for mtDNA and 39 samples for nuDNA, representing 33 of the ca. 40 currently recognized species (Barabanov & Ananjeva, 2007; Uetz & Hošek, 2016). The concatenated aligned mtDNA dataset encompassed 2,703 bp and the nuDNA dataset 1,488 bp. Information on the length of the fragments and variability were given in Table S3. Uncorrected mtDNA genetic distances within Phrynocephalus were given in Table 1 (below diagonal).

Table 1 Uncorrected p-distances for concatenated sequences of nuDNA (above diagonal) and mtDNA genes (below diagonal) (%) for species groups of Phrynocepahlus (1–11).

#	Group	1	2	3	4	5	6	7	8	9	10	11	
1	arabicus–maculatus	11.33/0.70	2.70	3.00	3.00	3.00	2.10	3.50	2.80	3.14	2.99	3.70	
2	interscapularis	17.71	11.45/1.57	3.40	3.60	3.60	2.50	3.60	3.40	3.40	3.39	3.81	
3	scutellatus	17.96	18.79	–	3.77	3.67	2.80	3.94	3.87	3.63	3.56	3.99	
4	ocellatus	17.08	17.24	18.54	7.09/1.00	1.52	2.50	1.57	2.90	1.73	1.56	2.00	
5	strauchi	16.45	16.28	18.40	11.92	–	2.40	1.60	2.82	1.57	1.53	2.00	
6	mystaceus	17.50	17.90	18.73	14.65	13.77	7.36/0.4	2.40	2.46	2.37	2.05	2.60	
7	helioscopus	18.08	17.68	18.99	14.81	14.14	15.64	11.09/1.14	2.92	1.77	1.67	2.07	
8	Oreosaura	16.47	16.08	18.09	14.26	13.59	15.04	14.80	7.93/0.75	2.85	2.50	3.16	
9	axillaris	16.91	16.79	18.24	14.53	12.72	13.98	13.47	13.52	2.18/–	1.08	1.75	
10	versicolor	16.89	16.89	18.12	14.65	13.21	15.78	15.03	13.65	13.64	6.95/0.31	0.60	
11	guttatus	16.92	17.01	18.27	14.40	13.10	15.16	14.91	13.69	13.48	9.28	5.84/0.58	
Note:

Values on the diagonal correspond to average uncorrected ingroup p-distances for mtDNA\nuDNA genes (%), respectively.

Phylogenetic inference from mtDNA

Analyses of the mtDNA data resulted in the majority of nodes receiving high BSP and BPP support. Topological patterns were in general congruent across analyses and the results of Solovyeva et al. (2014). The ML tree is shown in Fig. 2. The result appeared to be insensitive to exclusion/inclusion of distant non-agamid and non-agamine outgroups (see Figs. S2 and S3).

Figure 2 Mitochondrial genealogy of the genus Phrynocephalus on the base of 2,703 bp (partial COI, Cytb, ND2, ND4 sequences).

Node support values are given for ML BSP/MP BSP/BI BPP, respectively. Color marking of species groups corresponds to Fig. 3 and Fig. S4.

Phrynocephalus was unambiguously monophyletic in all analyses (Fig. 2). Several nodes in the mitochondrial tree appeared insufficiently resolved. Nevertheless all species of Phrynocephalus were consistently assigned to one of the ten strongly supported matrilines (for their distribution see Fig. S4): Subgenus Microphynocephalus, joining the small-sized, sand-dwelling Phrynocephalus from Middle Asia and the Middle East (Fig. 2; lineage A).

Subgenus Phrynosaurus represented by Phrynosaurus scutellatus from Iranian Plateau (Fig. 2, lineage B).

Near and Middle East Phrynocephalus: Phrynocephalus arabicus Anderson, 1894 and Phrynocephalus maculatus Anderson, 1872 (Fig. 2; lineage C), with the latter species being paraphyletic with respect to the former.

Subgenus Megalochilus Eichwald, 1831, including the large-sized, sand-dwelling Phrynocephalus mystaceus (Pallas, 1776) from Middle Asia (Fig. 2, lineage D).

Subgenus Oreosaura joining viviparous Tibetan species (Fig. 2, lineage E).

Middle Asian sun-watchers encompassing Phrynocephalus helioscopus and allied taxa (Fig. 2, lineage F; helioscopus-group).

Southern Middle Asian (Turan) Phrynocephalus raddei (Boettger, 1888), Phrynocephalus ocellatus, P. rossikowi, and Phrynocephalus strauchi Nikolsky, 1899 (Fig. 2, lineage G; raddei-group). P. rossikowi was omitted in the earlier mtDNA study of Solovyeva et al. (2014); our data strongly support its placement within the P. raddei species group.

Tibetan oviparous Phrynocephalus axillaris Blanford, 1875 (Fig. 2, lineage H).

Phrynocephalus versicolor species complex, inhabiting northern plains of Central Asia (Fig. 2, lineage I; versicolor-group). The versicolor-group had two sublineages: Phrynocephalus hispidus Bedriaga, 1909 from Mongolian Dzungaria and Phrynocephalus sp. 1 from Gansu; and Phrynocephalus przewalskii Strauch, 1876 + Phrynocephalus frontalis Strauch, 1876 + P. versicolor from central China and Mongolia joined with Phrynocephalus kulagini Bedriaga, 1909 from Tuva Republic (Russia).

Phrynocephalus guttatus species complex, widespread in plains of Kazakhstan and northern Caspian region (Fig. 2, lineage J; guttatus-group). Within the guttatus-group, P. guttatus, Phrynocephalus alpherakii Bedriaga, 1905 and Phrynocephalus moltschanovi Nikolsky, 1913 clustered together.

Phylogenetic inference from nuDNA and mito-nuclear discordance

Maximum likelihood, MP, and BI analyses of the concatenated nuclear DNA dataset resulted in highly congruent trees (Fig. 3). Exclusion of non-agamid taxa did not change the topology significantly (Fig. S5). Phylogenetic trees resulted from separate analyses of individual genes were shown in Figs. S6–S9; values of AU-tests for nuDNA genes compatibility were given in Table S8. The topology of the *BEAST species-tree for Phrynocephalus and the levels of nodal support (Fig. 4) coincided with the concatenated nuDNA dataset tree (Fig. 3) and were in good correspondence with the topologies from three of the four nuDNA genes (NKTR, RAG-1, and AKAP9).

Figure 3 Phylogenetic ML tree reconstructed from concatenated alignment of the nuclear genes RAG-1, BDNF, AKAP9 and NKTR.

Numbers on tree nodes indicate bootstrap values (BS) and posterior probabilities for ML BSP/MP BSP/BI BPP, respectively. Color marking of species groups corresponds to mitochondrial lineages; see Fig. 2 and Fig. S4. Thumbnails show representative species of each Phrynocephalus species group (to scale; note large size of P. mystaceus). Photographs by E. A. Dunayev and R. A. Nazarov.

Figure 4 Species tree reconstructed by *BEAST analysis with the nuclear genes RAG-1, BDNF, AKAP9, and NKTR.

Bayesian posterior probabilities (BI BPP) values are given only for strongly supported nodes. For Clades I–III definitions see “Discussion.”

Monophyly of Phrynocephalus received high support as did several species-groups: Microphrynocephalus (Fig. 3, lineage A: 99/100/1), Arabian species-group (Fig. 3, lineage C: 100/100/1), Megalochilus (Fig. 3, lineage D: 100/100/1), Oreosaura (Fig. 3, lineage E: 100/100/1), and P. raddei species-group (Fig. 3, lineage G: 95/84/1). The P. heliosopus-group obtained low support (Fig. 3, lineage F:–/–/0.89), but high support in the species-tree (Fig. S4). Monophyly of the clade containing the P. guttatus- and P. versicolor-groups was highly supported (Fig. 3, lineages I, J: 100/99/1), but interrelationships within this clade remained unresolved. The P. versicolor-group was paraphyletic with respect to the P. guttatus-group, though with some support only from ELW (Fig. 3: 90/–/–).

The nuDNA phylogeny of Phrynocephalus conflicted significantly (p < 0.05; AU) from the matrilineal genealogy. The nuDNA topology depicted three main clades (Fig. 3): (1) Microphrynocephalus, P. scutellatus, Arabian species-group and Megalochilus (clades A–D; 99/96/1); (2) Oreosaura (clade E; 100/100/1); and (3) all other Phrynocephalus (clades F–J; 100/99/1). Most notably, the placements of Megalochilus and Oreosaura differed in important ways. Matrilineally, Oreosaura and Megalochilus aligned with Middle and Central Asian “core Phrynocephalus” with strong support (99/88/1.0). In contrast, the nuDNA biparental phylogeny united Megalochilus with Arabian and Iranian species (75/90/0.99), including the P. arabicus-group, P. scutellatus and Microphrynocephalus with Oreosaura forming a sister-group to clades A–D. Other notable conflicts also occurred. The nuclear phylogeny did not depict a shared heritage for P. scutellatus and the Arabian species-group (Fig. 3), as did the mtDNA genealogy, but rather P. scutellatus (clade B) was the sister-lineage of Microphrynocephalus (Fig. 2). The phylogenetic position of P. strauchi was contentious; analyses of the nuDNA dataset did not group it with the mtDNA raddei-group. Similarly, the phylogeny placed P. axillaris as a sister-lineage of the P. guttatus–versicolor-group (Figs. 3 and 4; Fig. S5; 95/91/1), but its matrilineal relationships were unresolved (Fig. 2).

We performed additional AU tree-selection test to test for significant differences between matrilineal genealogy and the nuclear phylogeny, including whether one or both datasets rejected alternative placements of particular clades. The test evaluated the conflicting positions of P. mystaceus, P. strauchi, P. scutellatus, Oreosaura, the basal position of Arabian species and Arabian species + Microphrynocephalus. The matrilineal genealogy was forced to the nuclear dataset and vice versa. AU indexes for the basal position of Arabian species or Arabian species + P. scutellatus in the matrilineal genealogy were not statistically rejected by nuclear markers (p = 0.217, p = 0.277, respectively). The alternative nuclear hypotheses for the clades A–E and D–H were not rejected by mitochondrial data. The matrilineal position of P. mystaceus within the lineage (Helioscopus + P. axillaris + P. mystaceus) was rejected by nuclear markers (p = 0.000), and vice versa mitochondrial markers rejected the nuDNA resolution of P. mystaceus + Arabian species + Microphrynocephalus + P. scutellatus (p = 0.000). The AU test for P. strauchi occurring within the raddei-lineage was not rejected statistically by nuclear data (p = 0.752) and the mitochondrial data did not reject the nuDNA resolution of P. strauchi + helioscopus-group (p = 0.857). Existence of the matriline Oreosaura + “guttatus” + “versicolor” was rejected according to nuclear data (p = 0.000) and vice versa the mitochondrial dataset rejected position of Oreosaura within A–E (p = 0.000). Finally, we tested topologies of trees based on each nuclear marker against final topology. The BDNF dataset rejected the topology. In contrast, original RAG-1 topology was rejected. p-Values of the AU-tests of alternative topologies were summarized in Table S9.

Divergence times and rates of change

Chronograms from mtDNA and nuDNA data were presented in Figs. S10 and S11, respectively. Timing of the internal nodes was summarized in detail in Table S10. Estimated node-ages and the 95% highest posterior density (95% HPD) for the main nodes were detailed in Table S10. The mtDNA dataset provided older estimates of ages as compared to nuDNA. All analyses proposed that the ancestor of Phrynocephalus originated between the end of Oligocene and beginning of the Miocene (mtDNA: 33.2 Ma (26.4–39.7); nuDNA: 26.9 Ma (22.6–31.7)) and the basal radiation dated to the middle Miocene (mtDNA: 19.3 Ma (14.9–23.5); nuDNA 14.8 Ma (12.0–17.5)).

Lineage-through-time (LTT) plots gave a graphical representation of lineage-accumulation (Fig. S12). The mtDNA (Fig. S12A) and nuDNA (Fig. S12B) plots had similar shapes that were best described as being anti-sigmoidal characterized by three periods of constant rate. The first rate constant was separated from the second by a plateau and occurred before 14 Ma for mtDNA and before 11 Ma for nuDNA and the second plateau occurred after these dates. The third period started after 5 Ma, followed by a slight rate-shift in both plots.

Ancestral area, substrate niche, and body size evolution modeling

The ancestral areas and biogeographic processes (vicariance, dispersal, and colonization routes) reconstructed from nuDNA data were shown in Fig. 5. The most likely biogeographic scenario suggested that the Middle East plus the Turan area (ME-TU) was the most probable ancestral area for Phrynocephalus, thus supporting the hypothesis of a southern origin. Paleogeography of central Eurasia in Miocene–Pliocene was shown in Fig. 6.

Figure 5 Differentiation of Phrynocephalus: BEAST chronogram on the base of nuDNA dataset with results of ancestral area modeling in Lagrange.

GLB, “Gomphotherium-landbridge”; MMCT, middle Miocene thermal optimum; MMCT, middle Miocene climatic transition; PPCO, Pliocene–Pleistocene climate oscillations; AR, near East and Arabia, MI, Asia Minor and Transcaucasia; KZ, Kazakhstan, North Caspian and Ciscaucasian deserts; CA, Central Asia; TU, Turan; TI, Tibet; ME, Middle East. For biogeographic areas definitions see Supplemental Information 2 and Table S10. For paleogeographic reconstruction see Fig. 6. Node values correspond to estimated divergence times (in Ma). Icons illustrate vicariant events, area expansion and local extinctions, respectively. Black line on the inset shows modern range of Phrynocephalus. Red line corresponds to temperature change during the Cenozoic; climatogram from Zachos et al. (2001).

Figure 6 Paleogeography of Paratethys basin in late Cenozoic and the hypothetical scenario for Phrynocephalus.

Paleogeographic reconstructions are based on Rögl (1999); Popov et al. (2004); Popov et al. (2009) for early (A) and middle (B) Miocene and Pliocene (C). Question marks denote possible areas of distribution of the common ancestor of Phrynocephalus. GLB—”Gomphotherium-landbridge” between Arabian plate and Asian mainland (18–17 Ma). Red dotted line—possible range of Phrynocephalus; red arrows—possible dispersal routes; Latin numbers correspond to hypothetical distribution of main Phrynocephalus Clades I–III (see Discussion).

Results of habitat evolution modelling for Phrynocephalus were given in Fig. 7. All simulations of the possible evolution of substrate niches in Phrynocephalus suggested that the most likely substrate type for the ancestor consisted of soft substrate, i.e., loose sands with non-differentiated proluvial sediments, such as clay or gravel.

Figure 7 Evolution of habitat preference in the Agaminae including the genus Phrynocephalus.

See “Materials and Methods” and Table S5 for habitat data and Table S6 for step-matrix showing transition between substrate niche states. Agaminae outgroups, except for Xenagama, Trapelus, and Bufoniceps, inhabit large rocks and cliffs. Two lineages within the subfamily independently adapted to sandy habitats: the common ancestor of Bufoniceps and Trapelus and the ancestor of Phrynocephalus. Main groups within Phrynocephalus evolved adaptations to life on large loose sand dunes (P. arabicus, P. mystaceus and Microphrynocephalus), stony and gravel highland deserts (subgenus Oreosaura), and on clay substrates with gravel (P. helioscopus–P. raddei group).

Results of ancestral state reconstructions of maximum SVL evolution for each taxon were shown in Fig. 8. Accordingly, the common ancestor of Phrynocephalus was likely significantly smaller (size category 87–97 mm) than its sister taxa Laudakia, Stellagama, and Paralaudakia (ancestral size category 147–157 mm). Most species of Phrynocephalus have been found to be smaller than their common ancestor (size categories 47–77 mm); however, several lineages of Phrynocephalus have subsequently increased their body size (P. maculatus: 87–97 mm; P. mystaceus: 77–127 mm), while further miniaturization was suggested for the Phrynocephalus interscapularis-group (37–47 mm).

Figure 8 Body size evolution among Agaminae including genus Phrynocephalus.

See Table S5 for maximum SVL data. Color of branches corresponds to SVLmax (see legend). Rock-dwelling Laudakia s.l. are characterized by larger body size, while the common ancestor of Phrynocephalus was likely smaller than other Agaminae; sand-dwelling Microphrynocephalus (P. ornatus–P. interscapularis group) and Megalochilus (P. mystaceus group) represent the most miniaturized and the largest lineages within the genus, respectively.

Discussion

Phylogenetic relationships of Phrynocephalus

Supplemental Information 1 provides a brief review on history of phylogenetic studies of the genus Phrynocephalus.

Phylogenetic placement of Phrynocephalus. Our results are in accordance with previous studies on the higher phylogenetic relationships within the subfamily Agaminae (Macey et al., 2000; Guo & Wang, 2007; Melville et al., 2009) (Figs. 2 and 3). Sand-dwelling Bufoniceps laungwalaensis (Sharma, 1978) from the Thar Desert of India, which was originally described as a species of Phrynocephalus (Sharma, 1978), is the sister-lineage of the Middle Eastern–Middle Asian genus Trapelus; this corresponds with results of Macey et al. (2006) and Melville et al. (2009). The clade (Phrynocephalus + Laudakia s.l.) is poorly resolved; the old age of this radiation, which we estimate as early to mid-Oligocene (Figs. S10 and S11), likely complicates phylogenetic resolution. Future genome-scale studies are likely to recover these relationships. Regardless, the monophyly of Phrynocephalus (excluding B. laungwalaensis) is unambiguous in all analyses.

Phylogenetic relationships within Phrynocephalus. Analyses of nuclear gene exons (Fig. 3) resolve the phylogenetic relationships for most of the species-groups revealed by the matrilineal genealogy (Fig. 2), including the following: P. interscapularis-group (subgenus Microphrynocephalus; lineage A on mtDNA-genealogy), subgenus Oreosaura (lineage E), P. guttatus- and P. versicolor-groups (lineages J and I, respectively), P. mystaceus (subgenus Megalochilus; lineage D), P. helioscopus (lineage F), and P. arabicus–P. maculatus (lineage C). The P. guttatus- and P. versicolor-matrilines are sister-groups, although concatenated analyses of nuDNA data nest the former within the latter (Fig. 3) and the species-tree suggests sister-group relationships (Fig. 4). These are the youngest associations within Phrynocephalus. MtDNA analyses resolve close relationships between the P. helioscopus-group and P. axillaris, but analyses of nuDNA data resolve P. helioscopus-lineage + P. ocellatus-lineage with strong support and place P. axillaris as the sister clade to the P. guttatus + P. versicolor lineages.

Overall, the nuDNA phylogenetic trees are generally better resolved and show higher nodal support values than the mtDNA trees. The nuDNA phylogeny also shows higher congruence with traditional systematics of the genus. Owing to biparental inheritance, we prefer the nuDNA topology as the phylogeny of the genus Phrynocephalus (Fig. 3), which coincides well with the species-tree (Fig. 4). Further, the matrilineal genealogy appears to resolve the introgression of mitogenomes via interspecific hybridization (below). In general, Phrynocephalus has three main clades, but the relationships among them are not well-supported.

Clade I (Fig. 4) contains the P. interscapularis-group (subgenus Microphrynocephalus), P. arabicus–P. maculatus-group, P. scutellatus and P. mystaceus (subgenus Megalochilus). The phylogenetic relationships between these four groups are essentially unresolved. This groups includes oviparous species inhabiting Arabia, Near to Middle East, and Middle Asia. Many of them associate with sand dunes. Monophyly of the P. arabicus–maculatus group and Microphrynocephalus (P. interscapularis–Phrynocephalus ornatus Boulenger, 1887-group) has strong support.

Clade II (Fig. 4) includes viviparous species inhabiting high elevations of the QTP (subgenus Oreosaura). Phylogenetic relationships within this group remain unresolved with the exception of a sister-species relationship between Phrynocephalus theobaldi Blyth, 1863 and Phrynocephalus forsythii Anderson, 1872. In the concatenated analysis of nuDNA exons (Fig. 3), monophyly of the group that includes clade I and clade II receives strong support (75/90/0.99), although this node does not receive high support in the species-tree analysis (Fig. 4).

Clade III (Fig. 4), the “core” Phrynocephalus, contains all remaining taxa of the genus and consists of oviparous lowland species inhabiting arid areas of Central and Middle Asia and the Middle East. It includes two strongly supported groups: P. helioscopus–P. raddei and P. guttatus–P. versicolor. Monophyly of the P. helioscopus species complex and the P. ocellatus–P. raddei-group receives strong support. The phylogenetic position of P. strauchi remains unresolved, while P. axillaris appears as the sister-taxon of the P. guttatus–P. versicolor-group, which coincides with morphology; P. axillaris and members of the P. guttatus–P. versicolor species complexes share a number of morphological similarities, including having bright axillary spots possibly used in signaling communication (Fig. 3).

Our phylogeny has better resolution and wider taxon sampling than previously published mtDNA-based genealogies (Pang et al., 2003; Guo & Wang, 2007; Melville et al., 2009; Solovyeva et al., 2014) and the nuDNA-based phylogeny (1,200 bp fragment of RAG-1) of Melville et al. (2009). Our tree also differs from the morphological phylogeny of Arnold (1999), in which the Tibetan viviparous Oreosaura appear to be deeply nested within the core clade of Phrynocephalus and nests with the P. helioscopus-group, while P. mystaceus, P. arabicus, Phrynocephalus longicaudatus Haas, 1957 (as P. maculatus) and Microphrynocephalus occupy the basal position in the Phrynocephalus radiation. The potential explanation of these differences may be connected with habitat preferences; adaptation to types of substrate may drive convergent changes and morphology. Both Tibetan viviparous Oreosaura and P. helioscopus-group members occur mostly on solid (rocky or clay) substrates and are superficially similar in external morphology, sharing robust body habitus, shortened tail, and other features. At the same time, sand-dwelling Microphrynocephalus, P. mystaceus and P. arabicus and Bufoniceps share such characters as presence of enlarged scale fringes on toes, enlarged or keeled scales on throat, eyelids or jaws, flattened body and tail, which are considered to be adaptive for life on large wind-blown sand dunes. Arnold (1999) rooted his tree with Trapelus and Bufoniceps. However, convergent similarity between send-dwelling Phrynocephalus and Bufoniceps potentially could lead to the basal position of P. mystaceus.

Mito-nuclear discordance due to ancient hybridization. The main difference between mt- and nuDNA trees is the positions of P. mystaceus: the nuDNA-based topology strongly suggests that P. mystaceus as a sister-group with respect to the Middle-Eastern P. interscapularis-group (Microphrynocephalus), P. scutellatus and the Arabian P. arabicus–P. maculatus-group (Fig. 3). In contrast, the mtDNA genealogy unambiguously places P. mystaceus within core Phrynocephalus with P. axillaris being a possible sister-lineage (Fig. 2). The AU tests either rejects or do not provide statistical support for genomic compatibility (except as noted above; Table S8). Mitochondrial and nuclear genetic markers have yielded many conflicting geographic patterns (reviewed by Toews & Brelsford, 2012). Examples of mito-nuclear discordance in reptiles remain limited (McGuire et al., 2007; Zarza et al., 2011; Pedall et al., 2010; Renoult et al., 2009; Leache & Cole, 2007; Ng & Glor, 2011; Nguyen et al., 2017), perhaps because the discrepancy is rarely tested. Different mechanisms and rates of evolution for mt- and nuDNA may account for some observed topologic discrepancies. Fisher-Reid & Wiens (2011) evaluated 14 vertebrate clades for which both mtDNA and nuDNA data exist and reported that 30–60% of the nodes differed between trees from the two genomes. The results of our AU-tests suggest that topological differences between mtDNA and nuDNA hypotheses can be statistically significant. Thus, our analyses suggest that the combining of data from the two genomes should be avoided or done with caution without first testing for compatibility. Equally important, analyses based on combined datasets may hide biogeographic histories of studied taxa due to gene sorting, genetic recombination and gene flow of nuDNA (Nguyen et al., 2017).

Several processes, including incomplete lineage sorting or introgressive hybridization, may best explain mito-nuclear discordance (Toews & Brelsford, 2012). The deep divergence between P. mystaceus and core Phrynocephalus, as well as the unique morphology of this large-sized species differing from any other congener (Fig. 3), renders incomplete lineage sorting an unlikely scenario (McGuire et al., 2007). Thus, in case of P. mystaceus, an ancient introgression of mitochondrial genome due to interspecific hybridization best explains the discordance. According to mtDNA time-tree (Fig. S10) divergence between mitochondrial genomes of ancestral P. mystaceus and P. axillaris took place in late Miocene (around 10.4 Ma), whereas the basal differentiation of Phrynocephalus is estimated as the middle Miocene (around 14.8 Ma; Fig. S11) according to nuDNA data. Hence, the possible hybridization between P. mystaceus and P. axillaris ancestors occurred approximately 5 Ma after speciation, and this roughly corresponds to the level of divergence within the present-day P. guttatus–P. versicolor species complexes. This timing makes interspecific hybridization possible. Further, male-biased dispersal favors mitochondrial genome introgression (Toews & Brelsford, 2012) and Phrynocephalus appears to have male-mediated gene flow. Discordant breaks in mtDNA and nuDNA markers occur in at least for four species: Phrynocephalus vlangalii–Phrynocephalus putjatai Bedriaga, 1909-groups (Noble, Qi & Fu, 2010; Qi et al., 2013), and in the P. przewalskii–P. frontalis-groups (Urquhart, Wang & Fu, 2009). Other lineages may have the same pattern. P. mystaceus is the largest species of the genus and it shows adaptations typical of a psammophilous lifestyle including triangular scales forming fringes on toes, ridged subdigital lamellae, distinctly flattened body and tail, and a pair of unique cutaneous flaps at mouth corners with numerous spiny scales along flap edges. Similar spiny scales can occur in mouth corners of miniaturized psammophilous members of the P. interscapularis-group and Phrynocephalus euptilopus Alcock et Finn, “1896” 1897. These species are similar to P. mystaceus and occur in the Middle East and Middle Asia. Thus, our nuDNA phylogenetic hypothesis corresponds with life history, biogeographic, and morphological similarities between P. mystaceus and smaller psammophilous taxa of Phrynocephalus.

Taxonomic implications

The results of our phylogenetic analyses require some taxonomic changes within Phrynocephalus. These include recommendations on genus- and species-level taxonomy.

Generic taxonomy of Phrynocephalus. The significant morphological diversity of Phrynocephalus has been reflected in their generic taxonomy. First, Eichwald (1831) proposed the new name Megalochilus for the largest species of the genus, P. mystaceus. Ananjeva (1987) recognized the genus, but this was not accepted by subsequent researchers (Zhao & Adler, 1993; Dunayev, 1996b; Arnold, 1999). Sharma (1978) described P. laungwalaensis from Rajasthan (India), but Arnold (1992) removed it from Phrynocephalus and reassigned it to monotypic Bufoniceps, which he treated as a sister-clade of Phrynocephalus, Subsequent molecular analyses supported this arrangement (Macey et al., 2006; Melville et al., 2009). Based on the phylogenetic analysis by Pang et al. (2003), Barabanov & Ananjeva (2007) suggested recognizing the Tibetan viviparous species-group as subgenus Oreosaura. Because the phylogenetic relationships within oviparous species remained largely unknown, this subgeneric taxonomy was not largely accepted. Solovyeva et al. (2014) analyzed four mtDNA gene fragments and the morphological data of Arnold (1999) and suggested that small-bodied sand-dwelling Phrynocephalus, including the P. interscapularis-group and P. ornatus, constituted a morphologically and phylogenetically distinct group; they erected new subgenus Microphrynocephalus for it. However, our nuDNA-based phylogeny requires reconsideration of this arrangement.

Excluding B. laungwalaensis, nuDNA, mtDNA (herein), and morphological (Arnold, 1999) analyses indicate monophyly of Phrynocephalus. Thus, the splitting of Phrynocephalus into several genera is inadvisable. Further molecular studies are necessary to test for monophyly of Laudakia s.l. and its relationships with respect to Phrynocephalus.

The recognition of subgenera within Phrynocephalus is a matter of taste. The genus has three clearly defined clades that may warrant taxonomic recognition. The largest of these clades, or the “core” Phrynocephalus (clade III, Fig. 4), encompasses the majority of oviparous species and corresponds to the nominative subgenus Phrynocephalus s.s. clade II (Fig. 4) unites viviparous species of the QTP and is well-defined by morphology and life history traits; subgenus Oreosaura applies to it. Last, clade I (Fig. 4) joins a number of species from Arabia, Middle East and Middle Asia that have plesiomorphic character states (Arnold, 1999, including the P. arabicus–P. maculatus-group, P. interscapularis–P. ornatus-group, P. scutellatus and P. mystaceus. The P. interscapularis–P. ornatus-group, which was erected as subgenus Microphrynocephalus Dunayev, Solovyeva, Poyarkov, 2014 (Solovyeva et al., 2014) are miniaturized species adapted to life in aeolian sand habitat and they share a number of morphological synapomorphies (Arnold, 1999) and geographic coherence. The older name Phrynosaurus Fitzinger, 1843 is available. However, the phylogenetic position of its type species—P. olivieri Duméril et Bibron, 1837, which is now considered as a junior synonym of P. scutellatus (Olivier)—within the clade remains unclear. Finally, P. mystaceus, to which the name Megalochilus Eichwald, 1831, is available, forms a sister-group with respect to all other members of clade I. This species has a unique morphology and evolutionary history, which likely includes an episode of ancient intraspecific hybridization with introgression of its mitochondrial genome from clade III species.

Two alternative taxonomic decisions are possible: recognizing the whole of clade I as Megalochilus, or splitting it into a number of smaller taxa, including Megalochilus, Phrynosaurus, Microphrynocepahlus, and an unnamed taxon for the P. arabicus–P. maculatus species group. Our analyses lack samples from a number of Middle Eastern species, which likely fall into clade I, including P. ornatus ornatus, Phrynocephalus clarkorum Anderson et Leviton, 1967, Phrynocephalus lutensis Kamali & Anderson, 2015, Phrynocephalus luteoguttatus Boulenger, 1887 and, most importantly, large-sized and psammophilous P. euptilopus. Accordingly, we suggest that further taxon sampling and additional nuDNA-markers be evaluated before making subgeneric changes in the interest of maintaining taxonomy stability.

Taxonomic differentiation within species complexes. Our results indicate that in many cases the currently adopted taxonomy is incomplete and does not reflect the actual biodiversity within Phrynocephalus. We briefly review these cases and provide corresponding taxonomic recommendations.

Lineage A. Microphrynocephalus. This group of miniaturized species inhabit wind-blown sands of southern Middle Asia (Turan) and Middle East. Our analyses include P. interscapularis, Phrynocephalus sogdianus Chernov, 1948 and P. ornatus vindumi Golubev, 1998 only. Although unsampled, P. ornatus ornatus, P. clarkorum, P. luteoguttatus, P. lutensis most likely belong to this group based on morphological characters (Arnold, 1999; Kamali & Anderson, 2015). P. sogdianus was described from southern-most Tajikistan by Chernov (1948) as a subspecies of P. interscapularis (type locality “vicinity of the Pyandzh [=Panj] village”). Later, Sokolovsky (1975) raised it to be a full species. Our locality for P. sogdianus, Kurjalakum Sands, occurs approximately 50 km from the type locality and the Vakhsh River Valley separates it from the type locality. Our substantial divergence in mtDNA sequences (uncorrected genetic p-distance = 3.9–4.3% for COI) coincides with morphological and distributional differences and favors the recognition of P. sogdianus as a full species.

Lineage B. Phrynocephalus scutellatus. The phylogenetic position of this small-sized species, which inhabits clay or gravel deserts on the Iranian Plateau, remains unclear within clade I. The recent molecular study of Rahiamian et al. (2015) identified at least four highly divergent matrilines in southern and north-eastern Iran. Thus, P. scutellatus might consist of a species complex that requires reconsideration.

Lineage C. Arabian group. Our analyses include psammophilous P. arabicus s.l., which inhabits aeolian sand dunes from the Arabian Peninsula to westernmost Iran, and P. maculatus, which occurs on hard substrates and has two presently recognized subspecies: P. m. maculatus (Anderson) from the Iranian Plateau and P. m. longicaudatus from the Arabian Peninsula. Our sampling lacks Phrynocephalus golubewii Shenbrot et Semenov, 1990 from Turkmenistan and Phrynocephalus sakoi Melnikov, Melnikova, Nazarov, Al-Johany & Ananjeva, 2015 from Oman. Melnikov et al. (2014) revised P. arabicus complex, considered Phrynocephalus nejdensis Haas, 1957 and Phrynocephalus macropeltis Haas, 1957 as valid species, and described the population of P. arabicus s.l. from western Iran as the new species Phrynocephalus ahvazicus Melnikov, Melnikova, Nazarov, Rajabizadeh, Al-Johany, Amr & Ananjeva, 2014. However, this taxonomic action was based primarily on small to moderate genetic distances between these forms (p-distance 2.7–6.0%) and differences observed in coloration of living animals. Because Melnikov et al. (2014) did not include many morphological and meristic characters that serve to diagnose the species, these data must be taken with caution (Kamali & Anderson, 2015). Our sample from western Iran corresponds to P. ahvazicus of Melnikov et al. (2014). However, we tentatively assign it as P. arabicus s.l. pending a further re-assessment of the taxonomy of the P. arabicus species complex. Previously, Solovyeva et al. (2014) suggested that paraphyly of P. maculatus s.l. occurred with respect to P. arabicus. Our multilocus nuDNA-based phylogeny agrees with the matrilineal genealogy and indicates that P. longicaudatus from the Arabian Peninsula is a sister-taxon of P. arabicus (p-distance = 8.0% for COI). This differs from P. maculatus from Iran being reconstructed as a sister-taxon to the clade joining P. arabicus and P. longicaudatus. Based on the principle of monophyly, as well as genetic and distributional differences, we raise P. longicaudatus to full species status as P. longicaudatus (Haas, 1957) comb. et stat. nov.

Lineage D. Megalochilus. P. mystaceus represents a widespread species-complex that inhabits wind-blown sands and large sand dunes from north-eastern Iran to the Turan region, Middle Asia, and the Caspian Basin. While intraspecific taxonomy within P. mystaceus is in a state of flux, we report on two highly divergent lineages within this complex that were firstly revealed by mtDNA sequences (Solovyeva et al., 2014; p-distance = 6.3–6.5% for COI). Nuclear markers also reveal deep divergence between these two lineages, one of which is restricted to East Khorasan Province, Iran (P. mystaceus 2) and another occupies the rest of species range in Middle Asia (P. mystaceus 1). Thus, the taxonomy of this complex requires further study.

Lineage E. Oreosaura. This clade consists of viviparous highland species inhabiting deserts of the QTP and our analyses evaluate Phrynocephalus erythrurus, P. forsythii, P. putjatai, P. vlangalii, and P. theobaldi within the complex. The phylogenetic relationships among these species remain essentially unresolved. Pang et al. (2003), Jin, Brown & Liu (2008), Jin, Liu & Brown (2017), Jin & Liu (2010), Noble, Qi & Fu (2010), and Zhang et al. (2010) assessed their phylogenetic relationships and biogeography.

Lineage F. P. helioscopus–Phrynocephalus persicus-group. This group, which was until recently considered to be a widespread polytypic species P. helioscopus s.l., occurs in the montane deserts from western and northern Iran, Transcaucasia, the Turan Region, and Middle Asia to the Caspian Basin in the west and westernmost China and Mongolia in the east. P. helioscopus has a robust, tuberculate morphology and inhabits mostly hard substrates in clay or clay/gravel deserts. Previous phylogenetic analyses of mtDNA (COI) and nuDNA (interSINE-PCR) by Solovyeva et al. (2011) indicated the presence of two main clades within this complex: P. helioscopus complex (Middle Asia and adjacent territories) and P. persicus De Filippi, 1863 complex (Iran and Transcaucasia), both of which contained a number of highly divergent lineages. Subsequent analysis of morphological characters resulted in recognizing seven subspecies within P. helioscopus and three within P. persicus (Solovyeva, Dunayev & Poyarkov, 2012). Our phylogeny does not support monophyly of the P. helioscopus + P. persicus group (Fig. 2), although the species-tree does (Fig. 4). The P. helioscopus complex is a monophyletic unit. Deep divergences in both mtDNA and nuDNA genes occur between Phrynocephalus saidalievi Sattorov, 1981 from Ferghana Valley, Uzbekistan, and Tajikistan and P. h. helioscopus + P. h. varius Eichwald, 1831 (p = 12.0–12.6% in COI). Eremchenko & Panfilov (1999) proposed full species status for P. saidalievi based on differences in karyotype and our analyses strongly support this arrangement. Based on the principle of monophyly, along with the molecular and morphological analyses of Solovyeva et al. (2011), Solovyeva, Dunayev & Poyarkov (2012), we also recognize Phrynocephalus meridionalis Dunayev, Solovyeva, Poyarkov in Solovyeva, 2012 comb. et stat. nov. This species is the sister-taxon of P. saidalievi and was originally described as a subspecies of P. helioscopus from the Surkhandarya Region of southern Tajikistan. The species differs markedly in its mtDNA sequences (p = 10.0–10.6% in COI), nuclear markers and morphology (Solovyeva et al., 2011, Solovyeva, Dunayev & Poyarkov, 2012). Future studies can address further taxonomic reassignments within the P. helioscopus and P. persicus complexes.

Lineage G. P. raddei–P. ocellatus group. This group contains a number of species that have comparatively small distributions involving gravel or clay-gravel deserts of Turkmenistan, Uzbekistan, and eastern Tajikistan. Unfortunately, we were unable to obtain nuDNA sequences from our sample of P. rossikowi due to poor quality of the DNA from this sample; however, our mtDNA analysis strongly suggests that P. rossikowi is a member of this species group, and this is concordant with morphological similarity and ecology of this species, which prefer solid substrates (salines) (Solovyeva, 2017). Until recently, with the exception of P. rossikowi, two species were recognized: P. raddei Boettger, 1888 (subspecies P. raddei raddei Boettger, 1888 from Turkmenistan and P. raddei boettgeri Bedriaga, 1905 from Uzbekistan) and Phrynocephalus reticulatus Eichwald, 1831 (subspecies P. re. reticulatus Eichwald, 1831 from Uzbekistan, P. re. bannikovi Darevsky, Rustamov et Shammakov, 1976 from Turkmenistan and P. re. strauchi from Ferghana Valley in Uzbekistan and Tajikistan; Terentjev & Chernov, 1949; Wermuth, 1967, Bannikov et al., 1977; Barabanov & Ananjeva, 2007). Barabanov & Ananjeva (2007) considered P. boettgeri a junior synonym of P. raddei. Golubev (1991) examined the type specimens of Agama ocellata Lichtenstein in Eversmann, 1823 and demonstrated that it was the senior synonym of P. reticulatus Eichwald, 1831, which he considered as a subjective junior synonym of the former. Subsequently, the name P. ocellatus (Lichtenstein in Eversmann, 1823) was widely accepted for over 25 years (Dunayev, 1996b, 2008; Golubev et al., 1995; Manilo, 2000, 2001; Manilo & Golubev, 1993; Szczerbak, 1994; Solovyeva et al., 2014). Despite that, Barabanov & Ananjeva (2007: 56) proposed to protect the name P. re. reticulatus and in doing so violated the principle of priority (ICZN, 1999). Herein, we follow Golubev (1991) and use the name P. ocellatus in order to maintain nomenclatural stability and priority.

Our mtDNA analyses suggest monophyly of the P. raddei boettgeri–P. ocellatus group and suggest that P. strauchi is their sister-group. Their relationships within Phrynocephalus remain unresolved (Fig. 2). Concatenated (Fig. 3) and species-tree (Fig. 4) analyses of nuclear loci resolve paraphyly within the group. The phylogenetic position of P. strauchi is unresolved and P. raddei boettgeri–P. ocellatus form a well-supported sister-group relationships with P. helioscopus–P. persicus (lineage F). This clearly supports giving full-species status of P. strauchi as suggested by Dunayev (1995). Our sampling within this group is incomplete because we lack P. bannikovi and P. raddei raddei from Turkmenistan. Further studies are required to clarify phylogenetic relationships within this group.

Lineage H. P. axillaris. Oviparous P. axillaris from sand deserts of Taklimakan and adjacent parts of western China is a highly divergent lineage according to our concatenated (Fig. 3) and species-tree (Fig. 4) analyses of nuDNA-markers. It is the sister-taxon of lineages I–J (P. guttatus–P. versicolor-group). These lizards have a slender habitus, share habitat preferences and have bright red to blue axillary spots used in intraspecific communication (Fig. 3). Zhang et al. (2010) reported on the phylogeography of this species.

Lineages I–J. P. guttatus–P. versicolor group. Analyses of both mtDNA (Fig. 2) and nuDNA (Figs. 3 and 4) datasets suggest monophyly of the group. They are morphologically diverse oviparous species from lowland deserts of northern Middle and Central Asia. While analyses of the mtDNA data indicate reciprocal monophyly of P. guttatus and P. versicolor lineages, analyses of nuDNA data nest latter within the former. The P. guttatus-group (lineage J) inhabits various types of deserts ranging from sand dunes to gravel deserts. They occur in the Middle Asia from Caspian Basin to the westernmost China. Dunayev (1996a, 2009) assessed the taxonomy and distribution of lizards in Kazakhstan. NuDNA sequences fail to resolve relationships within this complex, yet mtDNA markers suggest the presence of three main matrilines: (1) P. guttatus, P. moltschanovi, and P. alpheraki from the Caspian and Aral basins and Ili Depression in eastern Kazakhstan; (2) Phrynocephalus kuschakewitschi Bedriaga, 1905 and P. incertus Bedriaga, 1905 from the Balkhash Lake Basin in eastern Kazakhstan; and (3) Phrynocephalus melanurus from the Zaysan Depression in northeastern Kazakhstan and Junggar Depression of northwestern China. Significant differentiation in mtDNA sequences (p=4.0–7.9% for COI), morphology and distribution argue for recognizing these forms of P. guttatus as full species. However, further research including more variable nuclear DNA-markers is necessary before doing so, especially in the Balkash Lake Basin in Eastern Kazakhstan, which cradles the highest species diversity of this group and where gene flow between species might take place (E. Solovyeva et al., 2018, Unpublished data).

Lineage I comprises the species of P. versicolor-group that inhabit lowland deserts of northern Central Asia. Whereas the nuDNA analyses do not resolve relationships within this group, mtDNA results suggest the presence of two matrilines, one containing populations from the Mongolian part of the Junggar Depression (Mongolian Jungaria), previously assigned to P. versicolor hispida (Orlova et al., 2014). Concatenated analysis of nuDNA loci suggest its recognition as the full species P. hispidus Bedriaga, 1909 comb. et stat. nov. The mtDNA divergence of this lineage from other members of P. versicolor group is substantial (p = 6.7–7.3%). Our analyses are in concordance the previous results by Gozdzik & Fu (2009) and Urquhart, Wang & Fu (2009) on genetic uniformity of P. przewalskii and P. frontalis. Despite the presence of two matrilines, the nuDNA markers do not differ between the two and P. przewalskii Strauch, 1876 appears to be the senior synonym for this taxon.

Phrynocephalus versicolor is a wide-ranging species that inhabits the Mongolian and Chinese Gobi Desert as far northwards as the Tuva Republic in southern Siberia (Russia). Traditionally, two subspecies were recognized: P. v. kulagini from Tuva and northern Mongolia and P. v. versicolor from the rest of species’ range. Previous studies (Wang & Fu, 2004) did not include samples of P. v. kulagini; our analyses strongly indicate paraphyly of P. versicolor s.l. with specimens from Tuva being significantly differentiated both in mtDNA (p = 5.18–5.37% for COI), nuDNA genes, and morphology. These results are supported by morphological differentiation (Dunayev, 2009), and this requires recognition of the full species P. kulagini Bedriaga, 1909 comb. et. stat. nov.

Implications for morphological and ecological evolution

Ancestral structural substrate niche. A debate exists on the ancestral habitat niche of Phrynocephalus. Several authors suggested that the common ancestor Phrynocephalus was likely adapted to soft, wind-blown sand dunes (Chernov, 1948; Whiteman, 1978; Arnold, 1999), whereas others argued that the group arose in stony or clay deserts with solid ground (Ananjeva & Tuniev, 1992; Golubev, 1989). Arnold (1999) provided a morphology-based phylogeny for Phrynocephalus, and with Trapelus and Bufoniceps reconstructed as its sister-taxa. He assumed that this group of genera demonstrated a gradual adaptation to soft substrates, such as loose aeolian sand-dunes.

We reconstructed the evolution of the preferred habitat types among all sampled Phrynocephalus and Agaminae outgroups using MP (Fig. 7; Table S6). Most Agaminae genera climb to some extent (Arnold, 1999) as Agama, Pseudotrapelus, and Laudakia s.l. exhibit. These taxa are found mostly frequently on large rocks and boulders, and this also occurs for most Trapelus, which inhabit sandy or gravel deserts, but eagerly climb bushes and trees. Only three genera—Xenagama, Bufoniceps, and Phrynocephalus—are strictly ground-dwelling, and this appears to be the derived condition. Our analysis suggests that two lineages within Agaminae independently adapted to life on soft substrates: the common ancestor of Trapelus and Bufoniceps, and the ancestor of Phrynocephalus.

Loose sands with non-differentiated proluvial sediments appear to be the ancestral habitat of Phrynocephalus. This initial adaptation could result in evolution of a set of advantageous features typical for this genus, such as (1) skin covering the tympanum, (2) lateral fringes of elongate scales on digits and eyelids, (3) countersunk jaws, (4) flattened body and tail, and (5) a characteristic burial behavior by lateral oscillations of the body (Arnold, 1999; Dunayev, 1996b). Further differentiation within Phrynocephalus likely led to adaptations of different lineages to contrasting habitat preferences. Phrynocephalus of clade I mostly specialize for life on sandy habitats such as large, wind-blown dunes (P. mystaceus and P. ornatus–P. interscapularis groups). Two species of clade I (P. scutellatus and P. maculatus) demark independent shifts to firmer ground (clay soils with gravel or salines). Clade II demonstrates adaptation to high elevation, stony or gravel deserts; a reversal to sand habitats occurs for P. forsythii, which inhabits the sandy Taklimakan Desert at lower elevations. Clade III likely had a sand-dwelling ancestor (Fig. 7). Many species in this group live on fixed sands with patches of gravel or clay, or they can be observed on various substrates. Some species switch to wind-blown dunes (P. axillaris, P. melanurus, P. przewalskii) or gravel (P. alpherakii, P. kuschakewitschi, P. melanurus, P. versicolor, P. strauchi). All members of the P. raddei–P. helioscopus group are specialized to life on hard substrates (clay, gravel, salines) and show no reversal to sand habitats, with the exception of P. helioscopus varius, which occurs in sandy areas of pine forests in the northernmost limit of its distribution in the Altai Region of Russia (Kotlov, 2008).

Body size evolution. Phrynocephalus show significant variation in body size ranging from 35–37 mm (P. ornatus vindumi) to 123 mm (P. mystaceus) (Table S6). The weighted squared-change parsimony algorithm serves to reconstruct the evolution of maximum SVL in Phrynocephalus and outgroup Agaminae (Fig. 8). Accordingly, rock-dwelling or climbing forms such as Agama, Trapelus, and Laudakia s.l. have larger SVLs than strictly ground-dwelling Bufoniceps, Xenagama, and Phrynocephalus. Body size decreases early in the history of Phrynocephalus, suggesting that initial miniaturization of its common ancestor was advantageous in wind-blown sand habitats. Strict sand-dwellers, Microphrynocepahlus (P. ornatus–P. interscapularis group) found in the aeolian sand dunes of the Middle East and Turan are the most miniaturized group of Phrynocephalus (SVLmax 47 mm). An example of an overt reversal in body size occurs in P. mystaceus, which is the largest member of the genus and is also specialized to large floating sand dunes. Similar change in body size possibly took place in psammophilous P. euptilopus, which appears to be closely related to P. interscapularis–P. luteoguttatus, but is much larger (SVLmax 63 mm) (Alcock & Finn, 1897; Arnold, 1999). Some species adapted to hard substrates also show an increase in body size, such as P. maculatus (SVLmax 91 mm) and P. putjatai (SVLmax 84 mm).

Historical biogeography

Time-tree and origin of Phrynocephalus. Estimates of divergence and diversification times for Phrynocephalus vary among authors. Macey et al. (1993) based on allozymes assumed that Phrynocepalus represented an ancient radiation and diverged about 35 Ma. According to immunological data, Ananjeva & Sokolova (1990) estimated the divergence of Phrynocephalus from other Agaminae took place around 11 Ma, while their allozyme data provided an estimate of 6 Ma. Using relaxed clock dating, Guo & Wang (2007) suggested a mid-Miocene origin (13.87 Ma, 95% CI [8.5–20.5]). Estimates by Melville et al. (2009) were older and varied for mtDNA and nuDNA datasets. Their mtDNA suggested all Phrynocephalus including P. interscapularis diverged 28.9 Ma (21.1–36.2 Ma). Excluding P. interscapularis, the estimated origin dated to 22.4 Ma (16.5–30.6 Ma) for mtDNA, and 15.8 Ma (11.8–23.0 Ma) for nuDNA.

Our analyses based on mtDNA data suggest that the ancestor of Phrynocephalus diverged from other Agaminae in early Oligocene around 33.2 Ma (19.92–45.69 Ma), or based on nuDNA data in late Oligocene around 26.9 Ma (22.44–31.27 Ma). The basal differentiation within the genus took place in early Miocene (mean 19.3 Ma; 95% CI [12.20–28.90] Ma) or mid-Miocene (mean 14.76 Ma; 95% CI [12.01–17.47] Ma) based on mtDNA and nuDNA data, respectively (Table S10). Due to absence of a reliable fossil record of Phrynocephalus, all of our calibration nodes correspond to relatively old splits between outgroup taxa, which can potentially result in biased date estimates (see, e.g., Brochu, 2004; Hugall, Foster & Lee, 2007).

Our estimates best fit the results of Melville et al. (2009), but are slightly younger. Melville et al. (2009) used one biogeographic and four fossil calibrations, including the enigmatic Bharatagama from Early–Middle Jurassic of peninsular India, which is interpreted as a putative stem acrodont (Evans, Prasad & Manhas, 2002). However, the attribution of this fossil is questionable (Townsend et al., 2011) and our calibration scheme does not include it (Table S5).

Biogeographic history of Phrynocephalus and Cenozoic climate change. Phrynocephalus is a characteristic element of the deserts of Palearctic Asia, and there is a substantial sympatry between species and species groups (Arnold, 1999) (Fig. S4). Several hypotheses have been invoked to explain the current broad distribution of the genus. Nikolsky (1916) assumed a Central Asian or Tibetan origin based on remarkable morphological diversity of Central Asian species (“northern origin” hypothesis). Ananjeva & Tuniev (1992) speculated that there were two original centers for the species of Phrynocephalus in the former USSR: Central Asia in the north and Middle Asia in the south. However, their complex hypothesis is not based on an estimated phylogeny and it omits numerous species of Phrynocephalus unique to Southwest Asia and China (Kamali & Anderson, 2015). Macey et al. (1993) suggested that the origins of Phrynocephalus trace back to Indian collision with Eurasia 35 Ma. Later Arnold (1999) suggested that Phrynocephalus evolved in the southern margins of the present distribution, i.e., in the Arabia–NW India area rather than in Central Asia (“southern origin” hypothesis). Some researchers (Wang & Macey, 1993; Zeng et al., 1997; Pang et al., 2003) argued that the basal differentiation of Phrynocephalus and the origin of the viviparous species group resulted from vicariance associated with the uplifting of the QTP. Guo & Wang (2007) suggested that Phrynocephalus diversified in the late Miocene to Pleistocene from centers of origin in temperate deserts of Central Asia, and the Tarim and Junggar basins. Several rapid speciation events followed this in a relatively short time (northern origin hypothesis). However, despite recent progress based on molecular phylogenetics (Pang et al., 2003; Melville et al., 2009; Guo & Wang, 2007), our understanding of the biogeography of Phrynocephalus, especially of the oviparous taxa inhabiting Middle Asia and South-Western Asia, remains very poor.

Phrynocephalus belongs to the subfamily Agaminae, which is hypothesized to have originated in Afro-Arabia and colonized Eurasia during a slow closure of the Tethys following movement of Arabian plate northwards (Macey et al., 2000). The landbridge connecting Afro-Arabia with Eurasia (known as “Gomphotherium-landbridge” or GLB; Figs. 5 and 6) was formed around 18–17 Ma and it facilitated a great faunal exchange between these landmasses (Rögl, 1998, 1999). After a temporary disruption, the landbridge has persisted continuously since the mid-Miocene about 15 Ma (Harzhauser et al., 2007; Pook et al., 2009; Metallinou et al., 2012; Scherler et al., 2013; Šmíd et al., 2013). Our analyses strongly suggest that the ancestor of Phrynocephalus diverged from other agamines around 27 Ma in late Oligocene, or ca. 9–10 million years predating the first connection of Afro-Arabia with Eurasia. Because Asian agamines are not monophyletic with respect to African taxa, if Macey et al.’s (2000, 2006) hypothesis on an Afro-Arabian origin of the subfamily is correct, then the ancestor of Phrynocephalus, Laudakia, and Trapelus + Bufoniceps should have colonized Asia independently from Arabia before formation of the GLB. Because Agamidae and putative Agaminae were present in Asia starting from the late Cretaceous and early Cenozoic (Estes, 1983; Borsuk-Białynicka & Moody, 1984; Alifanov, 1989; Borsuk-Białynicka, 1996; Prasad & Bajpai, 2008), we cannot exclude an Asian origin of Agaminae with subsequent colonization of Africa. The oldest known fossil of Phrynocephalus is reported from eastern Turkey (Zerova & Chkhikvadze, 1984); this record is quite young quite young (ca. 5 Ma) and not reliable. The driver of the basal differentiation within Agaminae in Oligocene (ca. 29 Ma) remains an enigma, as does the distribution of the common ancestor of Phrynocephalus (Fig. 5).

Figure 5 shows the main events in biogeographic history of Phrynocephalus inferred from molecular analyses and paleorange reconstructions. The basal radiation of Phrynocephalus happened around 14.8 Ma and likely took place in Middle East or southern Middle Asia (Turan) (Fig. 5). This scenario cannot reject the “southern origin” hypothesis for Phrynocephalus, as it does for the other hypotheses.

One of the most remarkable episodes of global climate evolution during the Cenozoic is the middle Miocene climatic transition (MMCT), which occurred between 14.8 and 14.1 Ma. (Flower & Kennett, 1994). At that time, a major and permanent cooling trend replaced the warm and humid tropical or subtropical climate of the mid-Miocene thermal optimum (17–15 Ma; Zachos et al., 2001; Böhme, 2003). The MMCT saw an increased meridional temperature gradient that strengthened the boundaries between climatic zones and increased aridification of the mid-latitudes (Flower & Kennett, 1994). The MMCT was synchronous with the Paratethys salinity crisis (Rögl, 1999), which was a major drying of the Paratethys Sea that may have formed significant wind-blown sand and evaporitic desert areas in Middle Asia. The MMCT aridification trend facilitated the spread of drier landscapes over the Mediterranean Basin, Arabia, the Iranian Plateau, and Middle Asia and promoted dispersal of xerophilic species (Manafzadeh, Salvo & Conti, 2014). The basal radiation of Phrynocephalus into clades I–III coincides perfectly with onset of the MMCT (14.8 Ma). The increasing aridification provided a diversity of desert habitats for Phrynocephalus to occupy as they started to disperse out of the ancestral area in Middle East and Turan (Fig. 5). A similar pattern was recently shown for agamid fan-throated lizards (genus Sitana Cuvier, 1829), which diversified in response to ongoing aridification of the Indian subcontinent in the late Miocene (Deepak & Karanth, 2018).

The ancestors of viviparous Oreosaura (clade II) supposedly colonized the QTP 13.5–10.0 Ma and diversification within this clade began around 3.8 Ma ago (Figs. 5 and 6). Our estimated divergence for Oreosaura is older than the earlier estimate of 9.7 Ma (95% interval: 7.2–13.0 Ma) by Jin & Brown (2013). Our date coincides with hypothesized major uplifting of the QTP (Shackleton & Chang, 1988) and, thus, is consistent with the view that viviparity evolved when this clade became restricted to regions of high elevation. The orogenesis of Tibet is traditionally regarded as the main driving force of Asian monsoons system and subsequent cooling and progressive aridization in Central Asia (Ramstein et al., 1997). However, recent works raise a doubt on the Miocene uplifting of the QTP, suggesting that Tibet has been 4–5 km high since the mid-Eocene (ca. 40 Ma), while Indian and Southeast Asian summer monsoons, and Central Asian winter monsoons arose at different times and are unrelated to Tibetan orogenesis (reviewed by Renner, 2016). These new data necessitate significant reconsideration of QTP historical biogeography and Phrynocephalus may represent a promising model group for such studies.

Oviparous clades I and III remained largely within the hypothetical ancestral range of Phrynocephalus (Middle East and Turan) (Figs. 5 and 6). Clade I has its highest diversity in the Iranian Plateau and adjacent parts of the Middle East. The ancestor of P. mystaceus diverged around 11.3 Ma; this large-sized psammophilous species spread northwards to Turan, Middle Asia and the Caspian Basin where vast areas of wind-blown sand deserts were formed after gradual drying up of the Paratethys Sea. Simultaneously, an ancestor of the P. arabicus–P. maculatus group occupied deserts of the Arabian Peninsula between 10 and 5 Ma (Fig. 5). The vicariant divergence between the ancestor of P. arabicus + P. longicaudatus from the Near East and Iranian P. maculatus happened around 4.8 Ma. This coincides with intensive uplifting of the Zagros Mountains in western Iran around 7 to 5 Ma ago (Mouthereau, Lacombe & Vergés, 2012).

Oviparous clade III encompasses the highest number of species and has the largest distribution in ranging from the Middle East and Asia Minor to Turan and Middle and Central Asia. Cladogenesis started in the late Miocene (ca. 8.9–7.1 Ma) and was likely influenced by progressive aridification of central Eurasia, orogenesis and changes in level of the Paratethys Sea. The age of the Central Asian deserts is questionable. The 22 Ma estimation of the onset of aridification in northwestern China (Xia & Hu, 1993; Guo et al., 2002) corresponds with estimates for the basal radiation in Dipodidae (Shenbrot et al., 2017), which is the major autochthonous component of Central Asian desert mammal community. The primary center of radiation for Phrynocephalus in Southwest Asia best explains their apparent lag in radiation.

The P. raddei–P. helioscopus group adapted to life on gravel or clay deserts. The highest diversity within this group occurs on alluvial plains of Middle Asia and Turan. The P. helioscopus–P. persicus species complex has largest distribution penetrating westwards to Transcaucasia and eastern Asia Minor, and northwards to the Caspian Basin, lowland deserts of Middle and Central Asia as far as western Mongolia (Fig. 5). Orogenetic processes in the Iranian Plateau and Kopet Dagh Mountains (Smit et al., 2013) possibly shaped the initial splits (6.2–3.4 Ma). The common ancestor of the P. axillaris and P. guttatus–P. versicolor group likely dispersed to Central Asia around 8.5 to 7 Ma, where it diversified. Accordingly, P. axillaris appears to have remained in the Taklimakan Basin and adjacent parts of Tibet, where desertification started from at least 22.6 Ma ago (Zheng et al., 2015). The ancestors of the P. guttatus–P. versicolor group penetrated to Middle Asia (Dunayev, 2009). Divergence between P. guttatus and P. versicolor species complexes occurred around 3.8 Ma, and this may coincide with accelerated uplifting of the Altai and Tianshan mountains around 5.0 to 3.1 Ma (Avouac et al., 1993; Abdrakhmatov et al., 1996; Charreau et al., 2005; Yuan et al., 2006). The ancestors of P. guttatus and P. moltschanovi further spread westwards, occupying the Caspian Basin and north-western Turan (Fig. 5).

Plio–Pleistocene glacial cycling likely profoundly affected subsequent radiations and range expansions within species complexes occupying northern parts of Middle and Central Asia (Aubekerov & Gorbunov, 1999). Formation of local montane glaciers or permafrost areas during glacial maximums could have led to the retreat of Phrynocephalus to warmer refugia followed by subsequent dispersals in warmer periods (Melville et al., 2009). Apparently, the QTP, now home to an impressive radiation of viviparous Oreosaura, was covered by a thick ice sheet in the Pleistocene (Kuhle, 1998). Hence, their distributions and the role they played in shaping the Central Asian biota remains insufficiently understood and requires further studies.

Conclusions

Exhaustive taxonomic sampling of Phrynocephalus is challenging. Some species of Phrynocephalus are only known from the type specimens and old collections (e.g., P. euptilopus and P. nasatus) (Barabanov & Ananjeva, 2007), while others occur in politically unstable zones (e.g., deserts of Pakistan, Afghanistan, and Taklamakan). Our analyses provide the most comprehensive taxonomic and gene sampling for Phrynocephalus to date. We evaluate 32 nominal taxa using four mtDNA and four nuDNA protein-coding genes. The sampling comprises over four-fifths of the species and covers the distribution of the genus. The mtDNA and nuDNA trees clarify the initial cladogenesis of these lizards. Statistically significant mito-nuclear discordance occurs likely due to hybridization and the introgression of mitogenomes. Analyses shed light on a number of taxonomic issues. Our results contribute to the interpretation of diversification patterns of Central Asian arid zone lizards and provides insights into the historical biogeography of this region.

Analyses confirm the monophyly of Phrynocephalus and infer its biogeographic history. The ancestral area of the Agaminae and factors that influenced its diversification remains uncertain. The origin of Phrynocephalus dates to the late Oligocene (26.9 Ma) and this precedes the formation of the mid-Miocene landbridge that connected Africa and Asia. The common ancestor of Phrynocephalus appears to have been a ground-dwelling, miniaturized agamine adapted to sand habitats. The basal divergence of Phrynocephalus into three major clades appears to have occurred in the Middle East or southern Middle Asia (Turan) around 14.8 Ma. This corresponds well with the mid-Miocene climatic transition—climate cooling that coincided with aridification and spreading of xerophytic plants across Mediterranean and Paratethys Basins. Subsequent drying up of the Paratethys Sea formed vast desert habitats that Phrynocephalus appears to have occupied. Two oviparous clades dispersed independently to lowland deserts of the Arabian Peninsula, Middle and Central Asia. Orogenetic processes and Paratethys Basin dynamics appear to have driven further cladogenesis, which Pliocene–Pleistocene climate oscillations built upon. Substantial variation in body size and morphology occurs in the oviparous lizards. Viviparous Oreosaura occupied the QTP around 13.5–10 Ma. Cladogenesis in this group dates between the late Oligocene and mid-Pliocene depending on the dataset (3.8 Ma from nuDNA, 6.4 Ma from mtDNA). This estimate coincides well with the divergence time of another viviparous group of lizards inhabiting Central Asia—the racerunner subgenus Pareremias (Lacertidae) (Orlova et al., 2017), which was dated to about 6.3 Ma from mtDNA data (Guo et al., 2011).

Climatic changes during the Cenozoic, including the ongoing aridification of central Eurasia, shaped the biodiversity of the region (Peng et al., 2006; Yang et al., 2006; Jin, Brown & Liu, 2008; Melville et al., 2009; Guo et al., 2011; Jin & Brown, 2013; Pisano et al., 2015). Most recent biogeographic studies assume the hypothesis that speciation in Central Asia correlated with the evolution of an East Asian monsoon climate triggered by the rapid uplifting of the QTP (Harrison et al., 1992, 1995; Zhisheng et al., 2001; Molnar, 2005). However, biogeographic histories of many taxa, including those inhabiting Central Asia and the QTP, might require reconsideration due to conflicting hypotheses on geological and climatic history of the region (Renner, 2016). Accordingly, our study highlights the importance of Cenozoic paleogeographic and paleoclimatic events in the diversification of Palaearctic lizards.

Supplemental Information

Supplemental Information 1 Review on history of phylogenetic studies of the genus Phrynocephalus.

Click here for additional data file.

Supplemental Information 2 Protocols of DNA PCR amplification used in this study.

Click here for additional data file.

Supplemental Information 3 Biogeographic area definition for Central Eurasia.

Click here for additional data file.

Supplemental Information 4 Divergence time estimated nodes (black) and calibration points (red) in Agamidae phylogeny for mtDNA (left) and nuDNA (right) topologies.

See Table S5 for calibrations and Table S10 for divergence time estimates.

Click here for additional data file.

Supplemental Information 5 Maximum likelihood inference tree of Phrynocephalus based on analysis of mtDNA dataset with all non-Agamidae outgroups excluded from the analyses.

ML BSP/BI BPP values are given for resolved nodes only.

Click here for additional data file.

Supplemental Information 6 Maximum likelihood inference tree of Phrynocephalus based on analysis of mtDNA dataset with all non-Agaminae outgroups excluded from the analyses.

ML BSP/BI BPP values are given for resolved nodes only.

Click here for additional data file.

Supplemental Information 7 Current distribution of matrilines, corresponding to species-groups and subgenera of Phrynocephalus.

Color marking of species groups corresponds to Figs. 2 and 3.

Click here for additional data file.

Supplemental Information 8 Maximum likelihood inference tree of the concatenated nuclear DNA data-set with all non-Agamid outgroups excluded from the analyses.

ML BSP/BI BPP values are given for resolved nodes only.

Click here for additional data file.

Supplemental Information 9 Maximum likelihood inference tree of Phrynocephalus based on analysis of nuclear AKAP9 gene fragment.

ML BSP/MP BSP values are given for strongly supported nodes.

Click here for additional data file.

Supplemental Information 10 Maximum likelihood inference tree of Phrynocephalus based on analysis of nuclear BDNF gene fragment.

ML BSP/MP BSP values are given for strongly supported nodes.

Click here for additional data file.

Supplemental Information 11 Maximum likelihood inference tree of Phrynocephalus based on analysis of nuclear NKTR gene fragment.

ML BSP/MP BSP values are given for strongly supported nodes.

Click here for additional data file.

Supplemental Information 12 Maximum likelihood inference tree of Phrynocephalus based on analysis of nuclear RAG1 gene fragment.

ML BSP/MP BSP values are given for strongly supported nodes.

Click here for additional data file.

Supplemental Information 13 Bayesian chronogram resulted from *BEAST analysis of the mitochondrial genes COI, Cytb, ND2 and ND4.

Node values correspond to estimated divergence times (in Ma). Grey-bars correspond to 95%-confidence intervals. Color marking of species groups corresponds to Figs. 2 and 3.

Click here for additional data file.

Supplemental Information 14 Bayesian chronogram produced by *BEAST analysis based on exons of the nuclear genes RAG-1, BDNF, AKAP9 and NKTR..

Node values correspond to estimated divergence times (in Ma). Grey-bar corresponds to 95%-credibility interval. Color marking of species groups corresponds to Figs. 2 and 3.

Click here for additional data file.

Supplemental Information 15 Lineage through time plots (LTT) of Phrynocephalus inferred from (A) mitochondrial DNA and (B) nuclear DNA datasets.

Click here for additional data file.

Supplemental Information 16 Geographic localities and museum voucher information of specimens used in this study.

ID/source–museum voucher/isolate ID; Locality–geographic locality of origin. No exact locality information is available for specimens obtained via pet trade.

Click here for additional data file.

Supplemental Information 17 Specimens used in this study and corresponding GenBank accession number information for four mtDNA and four nuDNA genes.

Specimen ID/source: Museum ID or GB–sequences downloaded from GenBank.

Click here for additional data file.

Supplemental Information 18 Characteristics of analyzed mtDNA and nuDNA sequences.

Total length (in b.p.), number of conservative (Cons.), variable (Var.) and parsimony-informative (Pars.-Inf.) sites are given (data presented only for the ingroup).

Click here for additional data file.

Supplemental Information 19 Primers used in this study.

“F,” “L”–forward primer, “R,” “H”–reverse primer.

Click here for additional data file.

Supplemental Information 20 Calibration points for divergence time estimation.

Node–tree node used for calibration, for node names see Fig. S10; Divergence time given in millions years (Ma); Fossil record–reference on the fossil record used for node calibration; Dataset for analysis–calibration used for mtDNA (mt), nuDNA (nu) or both (mt,nu) datasets.

Click here for additional data file.

Supplemental Information 21 Matrix of modern species distribution within Phrynocephalus, substrate niche and maximal body size data for Phrynocephalus and Agaminae outgroups.

Geographic regions: AR–Near East and Arabia, MI–Asia Minor and Transcaucasia, KZ–Kazakhstan, northern Caspian and Ciscaucasian deserts, CA–Central Asia, TU–Turan, TI–Tibet, ME–Middle East. Maximum SVL (maxSVL) values are given in mm. Substrate states (Substrate) are classified as: (1) loose sand dunes; (2) sands with non-differentiated proluvial sediments, e.g. gravel or clay; (3) gravel and stone deserts; (4) clay soils and salines; (5) clay soils mixed with gravel; (6) large rocks and cliffs.

Click here for additional data file.

Supplemental Information 22 Step-matrix showing transition between substrate niche states.

Numbers encode: (1) loose sand dunes; (2) sands with non-differentiated proluvial sediments, e.g. gravel or clay; (3) gravel and stone deserts; (4) clay soils and salines; (5) clay soils mixed with gravel; (6) large rocks and cliffs; see Fig. 7.

Click here for additional data file.

Supplemental Information 23 Values of AU-test for nuclear genes compatibility.

H0–test hypothesis (analysis as a separate marker), H1–original analysis (analysis within the concatenated alignment); AU–AU-test value.

Click here for additional data file.

Supplemental Information 24 Values of AU-tests for different topological hypotheses.

H0–test hypothesis, H1–original topology; nu_vs_mt–testing nuclear topology on mitochondrial dataset, mt_vs_nu–testing mitochondrial topology on nuclear dataset; AU–AU-test value.

Click here for additional data file.

Supplemental Information 25 Divergence time and ancestral area estimates.

For node names see Supplementary Figure S1. Estimated age is given in (Ma) for nuDNA and mtDNA datasets; results of ML biogeographic area reconstruction in Lagrange (Ancestral Area) is given for nuDNA topology only. For biogeographic areas definition see Fig. 6 and Supplementary File 2.

Click here for additional data file.

Supplemental Information 26 DNA alignment for AKAP9 nuDNA gene.

Click here for additional data file.

Supplemental Information 27 DNA alignment for BDNF nuDNA gene.

Click here for additional data file.

Supplemental Information 28 DNA alignment for NKTR nuDNA gene.

Click here for additional data file.

Supplemental Information 29 DNA alignment for RAG1 nuDNA gene.

Click here for additional data file.

Supplemental Information 30 DNA alignment for COI mtDNA gene.

Click here for additional data file.

Supplemental Information 31 DNA alignment for ND2 mtDNA gene.

Click here for additional data file.

Supplemental Information 32 DNA alignment for ND4 mtDNA gene.

Click here for additional data file.

Supplemental Information 33 DNA alignment for Cytb mtDNA gene.

Click here for additional data file.

The authors are grateful to the following colleagues who took part in the fieldwork, collected material and discussed the results, assembled and facilitated obtaining samples, as well as providing access to collections: V. F. Orlova, E. V. Vashetko, M. A. Chirikova, Weiwei Zhou, Wei Gao, T. N. Duysebayeva, P. V. Kvartalnov, T. A. Nazhmudinov, M. N. Yakubov, L. A. Neymark, A. A. Vedenin, I. V. Artyushin, and E. A. Peregontsev. NAP thanks Alexandra A. Elbakyan for help with accessing required literature. We are most grateful to Marcio Pie, Georgy Shenbrot and an anonymous reviewer for most useful comments on the earlier version of the manuscript.

Additional Information and Declarations

Competing Interests

Author Contributions

Animal Ethics

DNA Deposition

Data Availability

The authors declare that they have no competing interests.

Evgeniya N. Solovyeva conceived and designed the experiments, performed the experiments, analyzed the data, contributed reagents/materials/analysis tools, prepared figures and/or tables, authored or reviewed drafts of the paper, approved the final draft, discussion of the results.

Vladimir S. Lebedev conceived and designed the experiments, performed the experiments, analyzed the data, authored or reviewed drafts of the paper, approved the final draft, discussion of the results.

Evgeniy A. Dunayev conceived and designed the experiments, contributed reagents/materials/analysis tools, authored or reviewed drafts of the paper, approved the final draft, discussion of the results.

Roman A. Nazarov conceived and designed the experiments, contributed reagents/materials/analysis tools, prepared figures and/or tables, authored or reviewed drafts of the paper, approved the final draft, discussion of the results.

Anna A. Bannikova conceived and designed the experiments, authored or reviewed drafts of the paper, approved the final draft, discussion of the results.

Jing Che conceived and designed the experiments, contributed reagents/materials/analysis tools, authored or reviewed drafts of the paper, approved the final draft, discussion of the results.

Robert W. Murphy conceived and designed the experiments, authored or reviewed drafts of the paper, approved the final draft, discussion of the results.

Nikolay A. Poyarkov conceived and designed the experiments, analyzed the data, contributed reagents/materials/analysis tools, prepared figures and/or tables, authored or reviewed drafts of the paper, approved the final draft, discussion of the results.

The following information was supplied relating to ethical approvals (i.e., approving body and any reference numbers):

No experiments were conducted on living animals; only museum collections were used for this work.

The following information was supplied regarding the deposition of DNA sequences:

Sequences of AKAP9, BDNF, NKTR, and RAG-1 genes presented here are accessible via GenBank accession numbers KJ363400–KJ363439, KJ363514–KJ363551.

The following information was supplied regarding data availability:

Specimens examined in this study are deposited in herpetological collections of the following museums: Zoological Museum of Moscow University (ZMMU, Moscow, Russia);

Institute of Zoology and Parasitology, Uzbek Academy of Sciences (IZIP, Tashkent, Uzbekistan);

Kunming Institute of Zoology, Chinese Academy of Sciences (KIZ, Kunming, Yunnan, Peoples' Republic of China).

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
