# Peer review of "Cenozoic aridization in Central Eurasia shaped diversification of toad-headed agamas (Phrynocephalus; Agamidae, Reptilia)"

_PeerJ, doi:10.7717/peerj.4543_

## Round 0.1 · original submission · Major Revisions

Although both reviewers were enthusiastic about your paper, they raised a number of important issues that need to be addressed before the paper can be accepted - particularly Reviewer 1.

Reviewer 1 ·

Basic reporting

The writing requires some major tune-up. See more detailed comments below.

Experimental design

The design and execution are competent.

Validity of the findings

Conclusions are generally supported by the data and analysis.

Additional comments

This is a very interesting study of a very interesting group. The data are substantial and the analysis is competent. My main issue with this manuscript is its presentation. Overall, the manuscript reads like a draft, rather than a polished final product. Much of the manuscript needs to be trimmed, smoothed and tied up. I have two relatively large comments and many small ones.

Major comments

1. The results of the mtDNA. The authors may choose to either re-analyze the data (with some addition or change to Solovyeva et al’s data) and present the results as new or present them only in discussion for comparison (remove all related sections in Methods and Results). This would include Figure 2, which is exactly the same diagram from Solovyeva et al. (2014).

2. For most tables and figures (including supplementary ones), more detailed titles/captions and explanations are needed. These tables and figure should be self-content (fully understandable without referring to text). For example, Table S5. All abbreviation should be explained. Also, are these variable sites from the entire dataset (including the outgroups) or among only the ingroup members? This makes a huge difference because this study included a large number of outgroups. The variable and informative sites could be mostly among the outgroup members and provide no information regarding the relationships among the ingroup members, which is the primary goal of this study.

Other comments:

Line 88 and Line 130. Number of species and the use of species names. Although the Reptile Database was cited, about half of the names used in this study differ from these in the database. This creates confusion. The Database may not represent the most authoritative conclusion, but it does include most recent published results and is a good place to start. I would like to suggest that the authors used the names from the Database as much as possible, with some modifications and additions. For example, P. hispidus can be P. versicolor hispidus and P. kulagini can be P. versicolor kulagini.

Line 108-115. The paragraph is one long sentence. The authors could choose to either delete the paragraph entirely, or provide a brief review of the history of the phylogenetic studies of the genus (and move some of the discussion to here, e.g. line 406-445).

Line 165-180. Outgroup selection. The inclusion of all the non-agamid species may not be helpful in resolving the phylogeny of the genus Phrynocephalus. Although the inclusion may help in providing more calibration points and allowing some discussion on Agaminae, it my cause more harm to the analysis than the benefits. I would suggest removing all the distantly related outgroup and limiting discussion beyond the genus Phrynocephalus (e.g. line 446-467, see below).

Line 186-205. These three paragraphs need to be re-organized. They are confusing. ML and BI can use the same partition scheme. Dividing the alignment into 12 partitions may not be the best choice (line 197; could be over-parameterize the model); allow a computer program (e.g. the new PAUP or PartitionFinder) to determine the best partition scheme.

Line 233. Using calibration points from outgroup carries a risk; it is well known that such use has a tendency to over-estimate the time of divergence. Thus, in discussion, such tendency should be taken into consideration.

Line 241-243. May provide a brief justification of using a seven-region scheme.

Line 266-268. The rooting designation: Technically correct but conceptually wrong. Should be rooted by all outgroup members.

Line 269-271. These numbers are important and the numbers in Table S5 are different from the numbers presented here. In particular, how many variable/informative sites when only the ingroup members are included? When outgroup were included, the numbers could be dramatically increased but the majority of them could be uninformative among ingroup members.

Line 273-281. In the method section, there is no description about how the mt data were analyzed. I was having the impression that the tree from Solovyeva et al was used and the mt data was not re-analyzed. If the data were re-analyzed, a corresponding section should be provided in the method.

Line 282-320. The phylogeny (Fig 2) is exactly the published figure from Solovyeva et al (2014). None of these is new, and therefore, this section should be completely removed or minimized.

Line 322-324. What are the methodological differences between the concatenated analysis (ML, MP, BI) and the BEAST species-tree approach? What does the congruence between the sets of methods imply? Can you include the mtDNA data as the 5th loci in the species-tree approach?

Line 328-331. As I suggested earlier, the authors should restrain themselves from discussing issues beyond the genus Phrynocephalus (in this case, the monophyly of Agaminae), because the sampling is not designed for addressing these questions.

Line 341-356. The detailed description of the nuDNA tree should be combined with the last paragraph (line 332-339).

Line 376-383. I would like to see the confidence intervals being reported here (in addition to the means).

Line 406-445. Much of the historical review can be reduced (and move to the introduction section).

Line 446-467. The authors should restrain themselves from discussing issues beyond the monophyly of Phrynocephalus. This paper is about the phylogeny of Phrynocephalus, not Agamidae, for which the sample is very limited. Delete points 1, 2, and 3.

Line 508-525. Comparison to Arnold’s (1999) morphological analysis is probably important. In addition to describe differences, potential explanation of the differences should be provided. Comparison to mtDNA-based genealogies can probably be moved to next paragraph.

Line 515-525. This section is irrelevant to the main objective and should be removed.

Line 624-778. This part of taxonomic review is long the thorough, and I am not sure what to recommend. On one hand, this information is very useful for taxonomist; on the other hand, the detailed discussion distract from the main focus, the phylogeny of the genus. The authors probably should limit their discussion on the validity of various species. It is best evaluated with extensive samples at the population level. With the current sampling, it is better to focus on high taxonomic level. For example, the versicolor species complex (line 761-778). With only 6 sequences (Fig. 2), it is not suitable to discuss the validity of several species. Studies such as Urquhart et al (2009) with more than 170 sequences are better positioned to explore these topics.

Line 797. Habitat and body size evolution. Is there any evidence from fossil records to corroborate the arguments?

Line 832. Biogeography discussion. How large are the confidence intervals of divergence times? Be aware that geological events are also hypotheses and often have alternative explanations. For example, the Middle Miocene Climatic Transition. A quick read of Flower and Kennett (1994) find “The early stage of this climatic transition from ∼16 to 14.8 Ma…” and then “In the later stage from ∼14.8 to 12.9 Ma, …”. Clearly the transition is not as sharply defined as 14.8-14.1 Ma, rather, much longer.

Figure 2. This is exactly the same diagram from Solovyeva et al. (2014). It should not be presented as new results. Also, more explaination is required in the caption, for example, what are all the numbers? On the left, it should read “amino-acid sequences”.

Figure 3. Are these animal pictures proportional to their true size? Clade J, BI value: it is the only clade without a BI value. High BS value but no BI is unusual. Is it a topo? Groups G and I are not monophyletic. Why is sp. light blue? Shouldn’t be part of group J (orange)?

·

Basic reporting

The manuscript is clear and well-written.
Background information is provided and literature references are complete.
The article is good-structured, with clear figures and tables.

Experimental design

The research is original and is within aims and scope of the journal
Research questions are well defined and relevant. The research substantially fill existing gaps in the knowledge of Phrynocephalus phylogeny.
The research is performed with high technical and ethical standards.
Methods are described with sufficient details and information.

Validity of the findings

Results are novel and interesting for herpetologists and other biologists working in the area of evolution of desert biota of Central Asia.
Data are robust and results are statistically sound.
Conclusions are well-stated and are linked to original research questions.

Additional comments

I have few minor comments.
1. There is disagreement between abstract and body text concerning reconstructed substrate preferences of the common ancestor of Phrynocephalus. In the Abstract, it is written that “The common ancestor of Phrynocephalus probably preferred soft sand substrates”. However, in the body text this substrate is defines as “stabilized sand habitats with clay or gravel”. I would describe this type of substrate as loose non-differentiated proluvial sediments at mountain’s pediments. Please, correct.
2. The use of geographic term “Middle Asia”. In the western tradition, this term is not used, and this region is named “western part of Central Asia”. In the Russian geographic tradition the term “Middle Asia” is used for Turan Deserts and neighboring southern mountains (Tienshan, Pamiro-Alay) but not for Kazakhstan. Authors use this term in some parts of the text (Introduction) for Turan + Kazakhstan and in other parts (Material and Methods, Results, Discussion) for Kazakhstan only, what is not correct.
3. Areas used for biogeographic reconstruction (lines 235-246) are just listed, but not defined. Graphical presentation of these areas appears only in the Fig. 7. Please, define these areas in the text and change “Middle Asia” to “Kazakhstan” (see above).

---

## Round 0.2 · Minor Revisions

Both reviewers were pretty happy with the revised manuscript. However, they identified a few remaining details that need to be corrected.

Reviewer 1 ·

Basic reporting

Line 160-161. One sentence does not make a paragraph. Please merge this paragraph with the paragraph above.

Line 167-176. All the three paragraphs are single-sentence paragraph. Merge them into one paragraph.

Line 172 and elsewhere. Many comas (,) are omitted. Please pay attention and add. For example, line 172 after “inference”, line 179, after “subset”.

Line 180. Bootstrap does not evaluated “clade stability”. It evaluates “nodal support”.

Line 181. It is probably better to abbreviate “bootstrap” as “BSP” (for bootstrap proportion), as oppose to “BS”.

Line 201-203. Another one-sentence paragraph. Either merge it with others or break the sentence into two. There are many other instances of single-sentence paragraph. Please revise.

Line 204-207. Re-phrase. Nodes received BPP below 0.90 may be considered as “not well-supported”, but certainly are not “unresolved”.

Line 204 and elsewhere, consistency. Both “PP” and “BPP” are used for Bayesian posterior probability. Suggest using BPP.

Line 300. Awkward sentence. Sentence may read “Phylogenetic trees resulted from separate analyses of individual genes were shown…”

Line 376-377. Need 1-2 sentences to describe the results.

Line 406. Re-phrase. “higher stability”? Do you mean well-resolved and well-supported?

Line 411. Re-phrase. “not fully resolved”? Do you mean not well-supported? They are fully resolved on both trees.

Line 422. Awkward sentence. Sentence may read “…monophyly of the group that includes Clade I and Clade II receives….”

Experimental design

The data and analyses are adequate and competent.

Validity of the findings

Conclusions are convincing.

Additional comments

I have only a few comments and all of them are editorial.

·

Basic reporting

See my previous review

Experimental design

See my previous review

Validity of the findings

See my previous review

Additional comments

I am satisfied with author's corrections made in response to my previous notes.
The only small change that I can propose for the new version of the manuscript is the next:
The sentence "The arid belt of Central Eurasia ..." (lines 54-55) should be changed to "The eastern part of the Great Palearctic Desert Belt..." due to there is the single desert belt spanning from Atlantic coast of North Africa to north-eastern China.

---

## Round 0.3 · accepted · Accept

I believe that you properly addressed all of the (minor) issues indicated by the reviewers in the last round of reviews.